# Chimeric deubiquitinase engineering reveals structural basis for specific inhibition of the mitophagy regulator USP30

**Nafizul Haque Kazi** [1,2], **Nikolas Klink** [1,2], **Kai Gallant** [1,2], **Gian-Marvin Kipka** [1,2] **& Malte Gersch** [1,2] ✉

The mitochondrial deubiquitinase ubiquitin-specific protease (USP) 30 negatively regulates PINK1–parkin-driven mitophagy. Whether enhanced mitochondrial quality control through inhibition of USP30 can protect dopaminergic neurons is currently being explored in a clinical trial for Parkinson's disease. However, the molecular basis for specific inhibition of USP30 by small molecules has remained elusive. Here we report the crystal structure of human USP30 in complex with a specific inhibitor, enabled by chimeric protein engineering. Our study uncovers how the inhibitor extends into a cryptic pocket facilitated by a compound-induced conformation of the USP30 switching loop. Our work underscores the potential of exploring induced pockets and conformational dynamics to obtain deubiquitinase inhibitors and identifies residues facilitating specific inhibition of USP30. More broadly, we delineate a conceptual framework for specific USP deubiquitinase inhibition based on a common ligandability hotspot in the Leu73 ubiquitin binding site and on diverse compound extensions. Collectively, our work establishes a generalizable chimeric protein-engineering strategy to aid deubiquitinase crystallization and enables structure-based drug design with relevance to neurodegeneration.

Parkinson's disease (PD) is a prevalent neurodegenerative disorder, characterized by the progressive loss of dopaminergic neurons in the substantia nigra. A hereditary early-onset form of the disease, termed autosomal recessive juvenile parkinsonism, accounts for up to 10% of all patients and has been linked to somatic mutations in genes encoding the PINK1 kinase and the E3 ligase parkin among others[1]. Subsequent investigation of these proteins has guided the discovery of a selective autophagy mechanism for mitochondria termed mitophagy and uncovered the central role of dysfunctional mitochondrial quality control in neurons for PD pathogenesis[2]. In this pathway, PINK1 is stabilized on mitochondria with reduced membrane potential to phosphorylate ubiquitin conjugated to outer mitochondrial membrane proteins[3]. This in turn leads to the recruitment and activation of parkin, which,

together with the autophagy machinery, mediates the lysosomal degradation of damaged portions of the mitochondrial network[4–6]. Importantly, mitochondrial dysfunction has been strongly implicated in the etiology of both idiopathic and genetic forms of PD[7].

The post-translational modification of mitochondrial proteins with ubiquitin thus plays central roles by providing a substrate for PINK1 as well as by mediating autophagy[8,9]. Consequently, the mitochondrial deubiquitinase (DUB) ubiquitin-specific protease (USP)30 has emerged as a critical negative regulator of mitochondrial quality control due to its ability to remove ubiquitin from a subset of mitochondrial outer membrane proteins[10–15] (Fig. 1a). Notably, in addition to PINK1–parkin-driven mitophagy[11,16], USP30 also negatively regulates basal, that is, PINK1-dependent but parkin-independent

[1]Chemical Genomics Center, Max Planck Institute of Molecular Physiology, Dortmund, Germany. [2]Department of Chemistry and Chemical Biology, TU Dortmund University, Dortmund, Germany. ✉e-mail: malte.gersch@mpi-dortmund.mpg.de

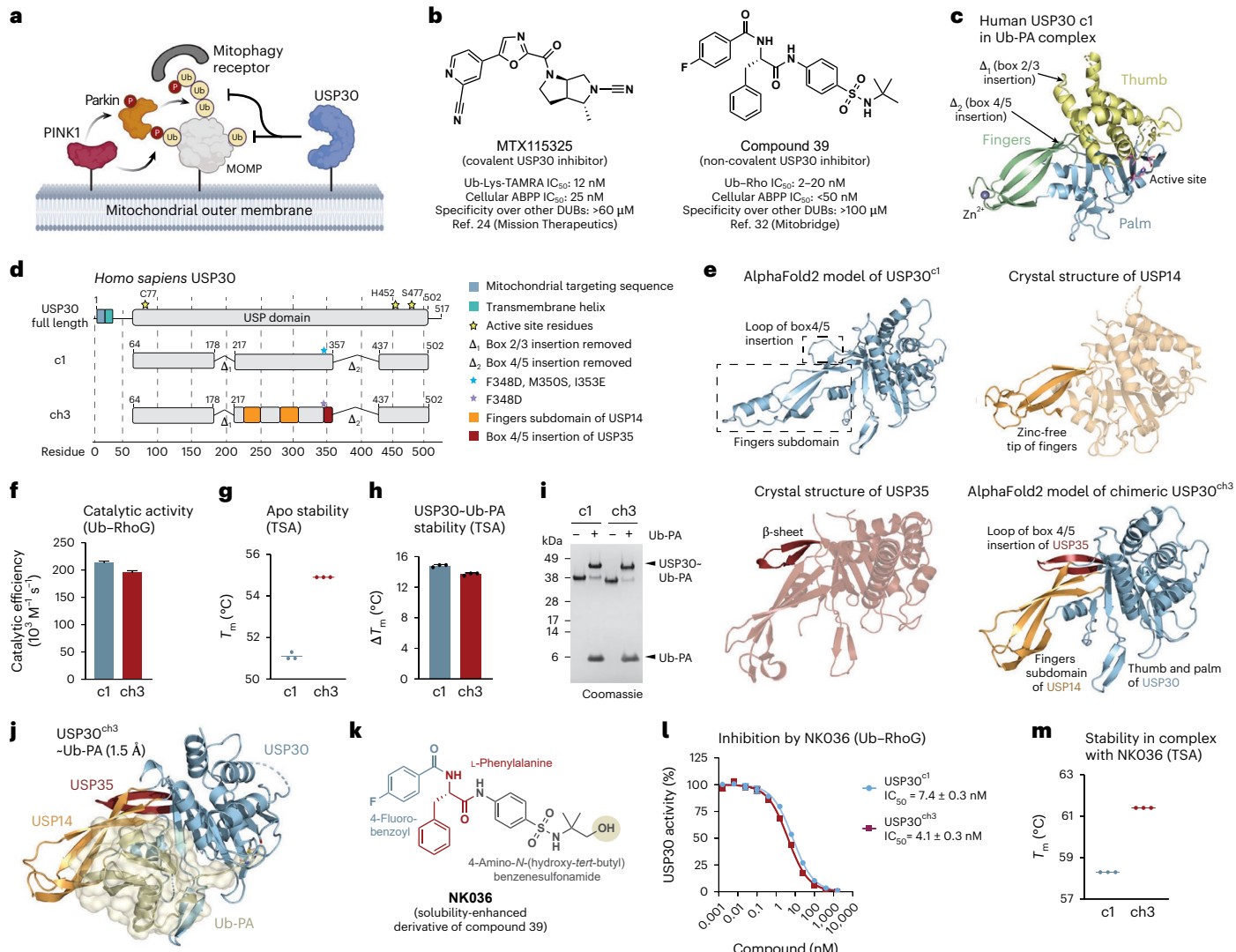

**Fig. 1 | Design, characterization and inhibition of chimeric USP30. a**, USP30 antagonizes PINK1–parkin-mediated mitophagy. MOMP, mitochondrial outer membrane protein. **b**, Chemical structures of a covalent and a non-covalent USP30 inhibitor. Inhibition of USP30 to enhance mitochondrial quality control is explored as a therapeutic strategy for Parkinson's and kidney diseases. ABPP, activity-based protein profiling. **c**, Crystal structure of human USP30 obtained with a previously engineered construct as the Ub-PA complex (PDB 5OHK). USP subdomains are shown in different colors. **d**, Architecture of full-length human USP30, the previously used c1 (named c13 in ref. 37) and the USP30 ch3 described here. See Extended Data Fig. 2 for other chimeras. **e**, AlphaFold2-predicted model of USP30 c1 (top left), crystal structures of the catalytic domains of USP14 (PDB 2AYN, top right) and USP35 (PDB 5TXK, bottom left) and AlphaFold2-predicted model of USP30 ch3 (bottom right). Regions used for grafting are shown in corresponding colors. **f**, Catalytic efficiencies of the indicated USP30 constructs, determined from Ub–RhoG cleavage assays. See Extended Data Fig. 3 for raw data. Mean ± s.e.m. (derived from curve fitting). **g**, Stability assessment of USP30 constructs in their apo states derived from thermal shift assays (TSAs). $T_m$, protein melting temperature. Mean ($n = 3$ independent replicates). **h**, Changes in protein stability upon binding to the ubiquitin probe Ub-PA. Mean ± s.d. ($n = 3$ independent replicates). **i**, Gel-based Ub-PA binding assay. **j**, Crystal structure of USP30 ch3-Ub-PA. Regions within the chimeric USP30 construct derived from different USP DUBs are shown in blue (USP30), orange (USP14) and red (USP35). The Ub-PA probe is shown in yellow. See Table 1 for statistics. **k**, Structure of NK036, a solubility-enhanced derivative of compound 39. **l**, Inhibitory potencies of NK036 for the indicated USP30 constructs. $IC_{50}$ values are given as mean ± s.e.m. (derived from curve fitting, with activity data for each concentration recorded as $n = 3$ independent replicates and shown as mean ± s.d.). **m**, Protein stability of the indicated USP30 constructs in the presence of NK036. Mean values are shown ($n = 3$ independent replicates). Panel **a** created with BioRender.com.

mitochondrial turnover[17,18]. Moreover, USP30 regulates mitochondrial protein import[19], mitochondrial morphology[20], mitochondrial abundance[21], apoptotic cell death[22] and pexophagy[17,23].

A fast-growing body of evidence suggests that USP30 is a highly promising drug target for PD, as its inhibition can protect dopaminergic neurons from α-synuclein-associated toxicity through increased levels of mitophagy[11,17,24]. This hypothesis has been underscored by studies in neuronal cell lines as well as in flies and in mice[18,24–28]. Inhibition of USP30 is also being explored as a therapeutic strategy in acute kidney injury due to links to mitochondrial dysfunction[29]. Several scaffolds of small-molecule USP30 inhibitors have been discovered[30,31] (Extended

Data Fig. 1), covering covalent and non-covalent modes of inhibition. Two compounds have recently been advanced into clinical trials by Mission Therapeutics.

The USP30 inhibitor compound 39 (Fig. 1b) displays half-maximum inhibitory concentration ($IC_{50}$) values against recombinant USP30 of 2–20 nM and cellular target engagement in the 10–50-nM range[27,32,33]. Moreover, the compound features pronounced specificity for USP30 both in cells and in vitro, with no other DUB being inhibited at 100 μM in a panel of recombinant enzymes, exceeding even the most specific covalent compound[24,27,33]. Compound 39 with a benzenesulfonamide scaffold[32,34] as well as the related naphthylsulfonamide MF-094 have

**Table 1 | Data collection and refinement statistics**

| | USP30ch3-Ub-PA (PDB 9F6G) | USP30ch3+NKO36 (PDB 9F19) |
|---|---|---|
| **Data collection** | | |
| Beamline | ESRF ID30A-3 | ESRF ID30A-3 |
| Wavelength (Å) | 0.9677 | 0.9677 |
| Space group | $P2_12_12_1$ | $P2_122_1$ |
| Cell dimensions | | |
| $a, b, c$ (Å) | 46.64, 50.71, 156.56 | 55.83, 73.84, 201.05 |
| $\alpha, \beta, \gamma$ (°) | 90, 90, 90 | 90, 90, 90 |
| Anisotropy correction | No | Yes |
| Observed reflections | 362,140 | 214,419 |
| Unique reflections | 58,606 | 15,438 |
| Resolution (Å) | 46.64–1.50 (1.53–1.50) | 59.52–2.75 (3.23–2.75) |
| Ellipsoidal resolution limits (Å) [direction] | – | 2.89 [a*] 3.53 [b*] 2.75 [c*] |
| $R_{merge}$ | 0.073 (1.240) | 0.166 (1.536) |
| $R_{meas}$ | 0.080 (1.350) | 0.173 (1.595) |
| $I/\sigma(I)$ | 10.7 (1.4) | 9.6 (1.9) |
| $CC_{1/2}$ | 0.998 (0.426) | 0.998 (0.730) |
| Spherical completeness (%) | 97.1 (96.8) | 68.7 (26.2) |
| Ellipsoidal completeness (%) | – | 91.9 (69.0) |
| Redundancy | 6.2 (6.4) | 13.9 (13.9) |
| Wilson $B$ (Å²) [direction] | 18 | 82 [a*] 161 [b*] 71 [c*] |
| **Refinement** | | |
| Copies (a.s.u.) | 1 | 2 |
| Resolution (Å) | 1.50 | 2.75 |
| Number of reflections | 58,467 | 15,431 |
| $R_{work}/R_{free}$ (%) | 18.3/21.1 | 21.8/26.6 |
| Number of atoms (non-hydrogen) | 3,126 | 3,932 |
| Protein | 2,767 | 3,791 |
| Ligand | 4 | 72 |
| Water | 355 | 69 |
| $B$ factors (Å²) | 29.0 | 79.2 |
| Protein (Å²) | 27.6 | 79.6 |
| Ligand (Å²) | 22.4 | 68.8 |
| Water (Å²) | 40.0 | 66.8 |
| RMSD | | |
| Bond lengths (Å) | 0.014 | 0.008 |
| Bond angles (°) | 1.33 | 1.05 |
| Ramachandran (favored/ allowed/outlier) (%) | 99.4/0.6/0.0 | 97.2/2.8/0.0 |
| Clashscore | 1 | 3 |
| Rotamer outliers (%) | 0.7 | 1.8 |

ESRF, European Synchrotron Radiation Facility; a.s.u., asymmetric unit.

been benchmarked in a range of cellular mitophagy assays[27,33,35,36]. A 'pseudo-covalent' binding mode has been proposed due to an exceptionally slow off-rate, and compound binding within the catalytic domain has been assessed by hydrogen–deuterium exchange mass spectrometry (HDX-MS)[33]. However, how such outstanding potency and specificity are achieved on the molecular level for any USP30 inhibitor scaffold is currently unknown, which has been hindering the acceleration of USP30 inhibitor development. This is likely related to the high flexibility and comparably poor crystallizability of the human USP30 protein, for which previously multiple rounds of construct optimization were required for structural characterization in ubiquitin-bound states[37] (Fig. 1c).

Here, we disclose a generalizable protein-engineering strategy based on USP DUB chimeras. By grafting structural elements of well-crystallizable human USP family members onto the periphery of the human USP30 protein, we generated stabilized and crystallizable protein constructs that retain DUB activity and the propensity to compound inhibition. By solving a high-resolution crystal structure of a suitably optimized USP30 construct in complex with a solubility-enhanced benzenesulfonamide inhibitor, we reveal the molecular basis for the highly potent and specific inhibition of USP30 by small molecules. This method not only elucidated the unique binding mode of this inhibitor class but also suggests a general strategy to enhance the crystallizability of other USP DUBs. We expand our analysis into a conceptual framework for specific USP DUB inhibition, which is based on a common ligandability hotspot in the Leu73 ubiquitin binding site and diverse compound extensions. Collectively, our work opens new avenues for the structure-based drug design of DUB inhibitors and enables the rational optimization of therapeutics targeting neurodegeneration.

## Results

### Design and characterization of chimeric USP30 constructs

Our work to understand the molecular basis of USP30 inhibition started with an engineered construct of human USP30 (construct 1 (c1)), for which crystal structures of ubiquitin-bound complexes were previously obtained[37]. This construct contains the three subdomains of human USP30 (palm, thumb and fingers) but lacks two largely unstructured sequence insertions[38,39] (Fig. 1c,d and Extended Data Fig. 2a). However, despite extensive co-crystallization screening with several potent small-molecule USP30 inhibitors, no initial crystals were obtained. We hence set out to enhance the crystallizability of the USP30 protein through further construct optimization. We were guided in our approach by a systematic review of all crystal structures of human USP DUBs deposited in the Protein Data Bank (PDB) (Supplementary Table 1). Of the 55 human USP enzymes, 21 have crystal structures of catalytic domains available (38%)[40,41]. Notably, eight of the 55 USP enzymes lack zinc-coordinating cysteine and histidine residues at the tip of their finger subdomain. Within this subset, six of the eight enzymes have been crystallized (75%)[39] (Extended Data Fig. 2b). This discrepancy is even more pronounced when focusing on the 14 human USP DUBs for which apo or inhibitor-bound structures were reported (25%). These include the same six DUBs that do not bind zinc (Extended Data Fig. 2b). These data suggest that USP DUBs lacking a zinc in their finger subdomain may have a higher propensity for crystallization. A focus on the fingers region for construct optimization was further motivated by (1) previous HDX-MS analysis suggesting enhanced flexibility compared to other parts of the protein and (2) HDX-MS-based mapping of the compound 39 binding site into the palm and thumb subdomains[33,37].

We envisaged that the generation of chimeric USP30 catalytic domains, in which the USP30 finger subdomain is replaced by sequences of equivalent regions in other USP DUBs, may facilitate crystallization (Fig. 1d,e). We focused on the finger domains of USP7 and USP14, which lack zinc ions and have both been crystallized in multiple apo and inhibitor-bound forms[42–48] (Extended Data Fig. 2c–f). We also included the USP DUB CYLD, which features truncated, zinc-free fingers[49,50] (Extended Data Fig. 2g), and also designed construct 2 (c2) in which the entire fingers were replaced by Gly–Ser linkers (Extended Data

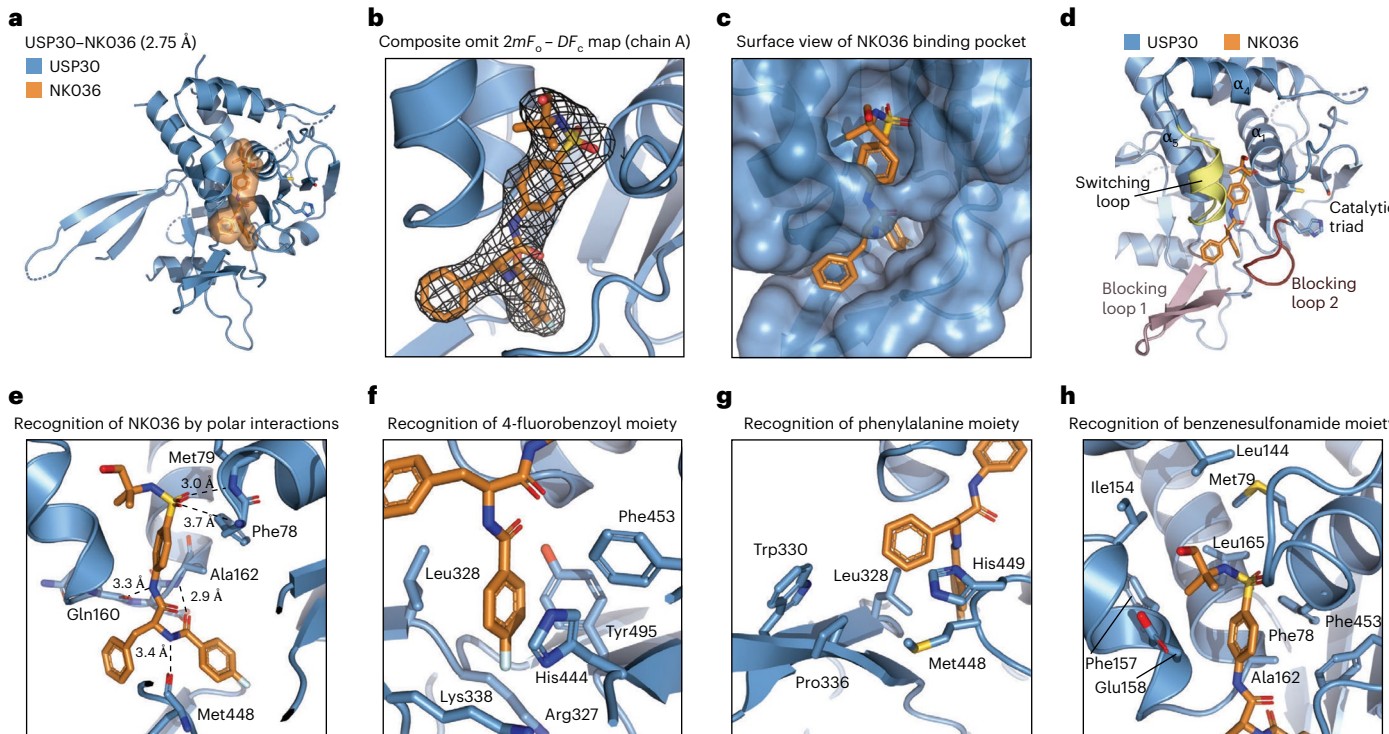

**Fig. 2 | Structure of USP30 in complex with NK036. a**, Cartoon representation of the crystal structure of USP30$^{ch3}$ bound to NK036. The compound is shown under an orange surface. **b**, Composite omit electron density map of NK036 in chain A ($2mF_o - DF_c$, contoured at $1\sigma$, covering all atoms of the compound, created with simulated annealing from the final coordinates). See Extended Data Fig. 5e,f for unbiased $mF_o - DF_c$ maps. **c**, Structure as in **a** with surface representation of USP30. **d**, Compound binding site highlighting typical USP regions involved in binding to NK036. These include the switching loop (yellow), blocking loop 1 (pink) and blocking loop 2 (red). Residues of the catalytic triad are shown as sticks. **e**, Close-up view of the compound binding site highlighting key residues involved in hydrogen bonding. **f**, Close-up view of the USP30 hydrophobic patch engaging the fluorobenzoyl moiety of the compound. **g**, Close-up view of hydrophobic interactions of the phenylalanine group of the compound. **h**, Close-up view of the benzenesulfonamide moiety of the compound engaged by USP30 residues.

Fig. 2h). In addition, we noticed that the loop of the box 4–5 insertion deletion features high flexibility in HDX-MS and elevated $B$ factors in ubiquitin-bound complexes. We planned a replacement by the equivalent region of USP35, as USP35 is the only DUB featuring secondary structure in this region (a short, antiparallel β-sheet)[51] (Extended Data Fig. 2f).

In an iterative process, we explored 15 chimeric constructs starting with boundary design by structure superposition, design validation by AlphaFold2-based modeling (Extended Data Fig. 2h–l), cloning (Extended Data Fig. 2m), protein purification and biophysical characterization (Extended Data Fig. 3a–e). This process is illustrated with four diverse chimeras: chimera 1 (USP30$^{ch1}$) features the fingers of USP7, chimera 2 (USP30$^{ch2}$) features the fingers of USP14, chimera 3 (USP30$^{ch3}$) features the fingers of USP14 and the box 4–5 insertion of USP35, and chimera 4 (USP30$^{ch4}$) features the fingers of CYLD (Extended Data Fig. 2i–l). All proteins were expressed and purified, and their stability was assessed in thermal shift assays. While inclusion of the USP14 and CYLD fingers did not alter protein stability, USP7 fingers destabilized the chimeric protein, whereas inclusion of the structured box 4–5 insertion of USP35 increased protein stability (Extended Data Fig. 3a). Chimeras 1–3 showed complete binding to the ubiquitin probe Ub-PA (Extended Data Fig. 3b), which correlated with protein stabilization (Extended Data Fig. 3c), whereas this was not the case for constructs with truncated fingers. Consistently, chimeras 1–3 showed high catalytic activity toward the fluorogenic substrate ubiquitin–RhoG (Ub–RhoG; Extended Data Fig. 3d,e), whereas USP30$^{ch4}$ and c2 were virtually inactive. These results are consistent with the large protein interaction surface contributed by the fingers for ubiquitin recognition. Importantly, chimeras 1–3 retained their propensity to be inhibited by compound 39, with IC$_{50}$ values between 0.3 and 0.7 nM compared

to 0.8 nM for construct 1 (Extended Data Fig. 3f,g). To assess inhibitor binding in all proteins, including the catalytically inactive chimeras, we measured inhibitor-induced changes in protein stability. The presence of compound 39 increased protein stabilities between 7 and 9 °C for all samples, which demonstrates that all USP30 constructs retain affinity for the benzenesulfonamide scaffold (Extended Data Fig. 3h).

**Inhibition of a chimeric USP30–(USP14–USP35) construct**
Upon surveying the collected data (Supplementary Table 2), we decided to focus on USP30$^{ch3}$. This construct containing the USP14 fingers and the USP35 box 4–5 insertion (Fig. 1d,e) features unaltered catalytic activity (Fig. 1f), displays an increased protein stability by approximately 4 °C (Fig. 1g) and reacts readily with Ub-PA (Fig. 1h,i). To validate our design, we solved the crystal structure of the covalent USP30$^{ch3}$-Ub-PA complex (Fig. 1j and Table 1). Both a rather high rate of initial crystal hits in coarse-screening plates and the high resolution of 1.5 Å without crystallization fine screening supported the hypothesis of increased crystallizability of this chimeric protein. Inspection of the electron density revealed that the boundary design allowed seamless chimeric sequence transitions and did not perturb the USP fold (Extended Data Fig. 4a). Consistently, the structure showed a highly similar arrangement compared to Ub-PA complexes of USP30, USP14 and USP35 (Extended Data Fig. 4b).

During the initial crystallization experiments, we noticed that compound 39 possesses poor solubility in aqueous buffers. We therefore synthesized NK036 as a solubility-enhanced derivative (Fig. 1k). This compound shares the 4-fluorobenzoyl group, the central L-phenylalanine and the benzenesulfonamide with compound 39 and features an additional hydroxyl group at the *tert*-butyl group. While this

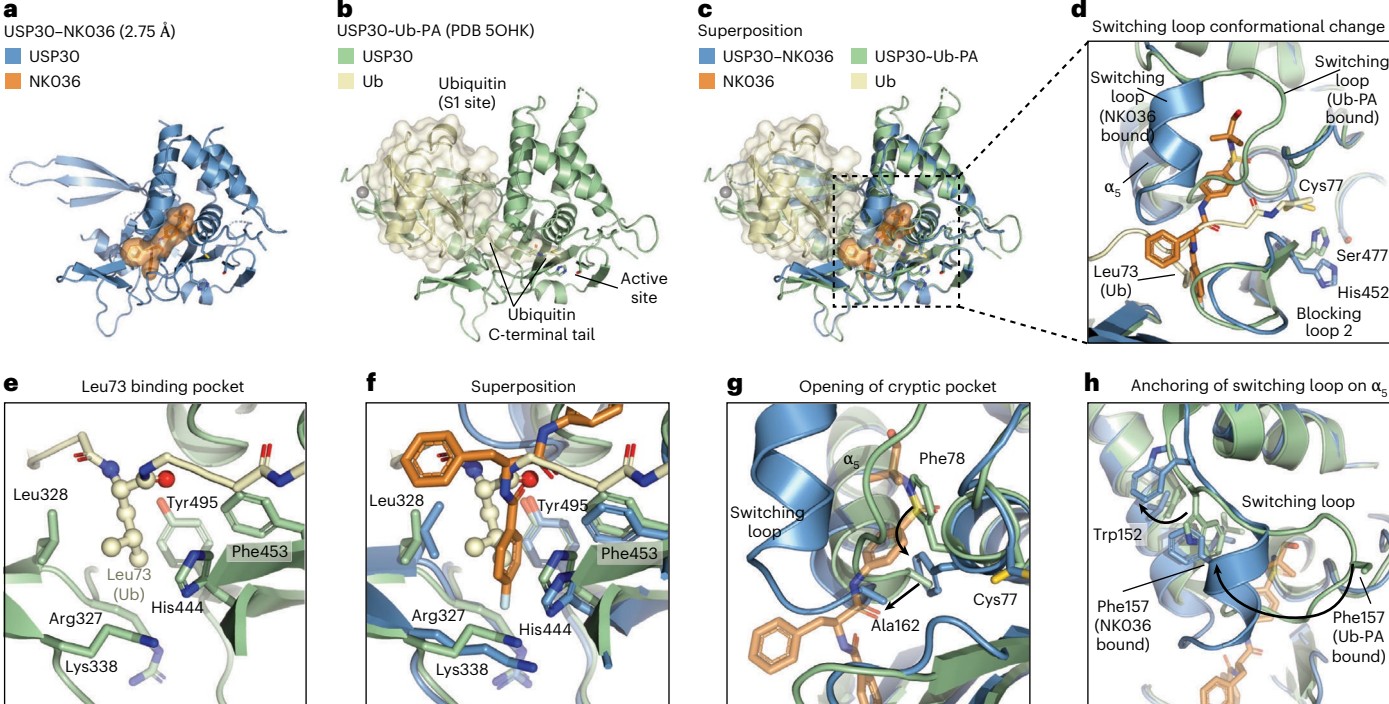

**Fig. 3 | Engagement of the Leu73 pocket and conformational plasticity of the switching loop underlie ligand engagement by USP30. a,** Cartoon representation of the crystal structure of USP30 bound to NK036. The compound is shown under an orange surface. **b,** Structure of the USP30-Ub-PA complex (PDB 5OHK). Ubiquitin is shown under a yellow surface. **c,** Superposition of **a** and **b**. **d,** Close-up view of the compound binding site. Catalytic triad residues of USP30 and Leu73 of ubiquitin are labeled. The conformational change of the USP30 switching loop is indicated. **e,** Close-up view on the engagement of the ubiquitin Leu73 side chain by USP30, with residues forming the hydrophobic pocket highlighted. **f,** Superposition of the structures, focused on the Leu73 binding pocket, showing its occupation by the fluorobenzoyl group of NK036. **g,h,** Close-up views of the conformational changes of the switching loop, focusing on entry of the cryptic pocket within the thumb subdomain (**g**) and anchoring of the switching loop on the $\alpha_5$ helix (**h**). Putative movements of residues are indicated with arrows. See also Supplementary Video 1, in which the equivalent transition is shown based on the Lys6 diubiquitin-bound structure of USP30 (PDB 5OHP)[37].

addition decreased potency by about one order of magnitude (with $IC_{50}$ values between 4 and 7 nM; Extended Data Fig. 3f,g), it improved NK036 solubility in aqueous buffers. Importantly, NK036 inhibits USP30[c1] and USP30[ch3] to a similar degree (Fig. 1l) and increases protein stability to above 61 °C (Fig. 1m), with the same relative stabilization of all constructs (Extended Data Fig. 3h). These data underscore the suitability of the stabilized USP30[ch3] construct and of NK036 as a potent USP30 inhibitor for co-crystallization studies.

**Structure of USP30 in complex with a non-covalent inhibitor**

The structure of USP30[ch3] in complex with NK036 was solved to 2.75 Å following crystal optimization through fine screening and additive screening as well as data optimization through multicrystal averaging and anisotropic scaling (Fig. 2a, Table 1 and Supplementary Table 3; see Methods for details). The asymmetric unit contained two copies of the protein–ligand complex. These could be superimposed with a $C_\alpha$-root mean square deviation (RMSD) of 0.4 Å (Extended Data Fig. 5a) and showed near-identical ligand geometries (Extended Data Fig. 5b). Both chains displayed clear electron density, which allowed unambiguous positioning of the inhibitor (Fig. 2b and Extended Data Fig. 5c–g). Chain A contained more ordered residues near the finger subdomain, which were partially disordered in chain B, whereas conversely chain B showed ordered loops on the opposite side near the thumb subdomain. These differences could be attributed to crystal contacts, and these regions are far away from the ligand binding site. Thus, chain A was used for all further analysis. The ligand was surrounded only by USP30-encoded side chains as well as secondary structures generated by USP30 residues, with chimeric portions of the sequence spatially separated (Extended Data Fig. 5h). This indicates that the chimeric

engineering in USP30[ch3] does not prevent the deduction of a bona fide USP30 inhibition mechanism.

NK036 was found to be engaged by both the palm and thumb subdomains (Extended Data Fig. 5i) and extensively surrounded by protein residues (Fig. 2c and Extended Data Fig. 5j). The phenylsulfonyl and *tert*-butyl moieties are engaged by the thumb subdomain, being buttressed between the switching loop, the $\alpha_5$ helix and the $\alpha_1$ helix. The phenylalanine and fluorophenyl moieties of the ligand are mainly surrounded by blocking loops 1 and 2 within the palm region (Fig. 2d). The two central amide bonds of the ligand are contacted through a total of three hydrogen bonds by the protein (Fig. 2e): the backbone carbonyl group of Gln160 engages the anilinic NH proton, the carbonyl group of Met448 contacts the other amidic NH proton, and the carbonyl group of the fluorobenzoyl moiety engages the Ala162 nitrogen. These polar interactions give rise to a star-like tripartite geometry of NK036 with the hydrophobic portions of the ligand extending into three separate areas:

1. The fluorobenzoyl ring binds into a hydrophobic pocket formed by Leu328, Tyr495 and Phe453 as well as aliphatic portions of the Arg327 and Lys338 side chains (Fig. 2f). While the imidazole ring of His444 binds the fluorophenyl ring through parallel π–π stacking, the Leu328 side chain on the other side of the ring creates a pin to close the pocket toward blocking loop 1.

2. The central phenyl ring of the ligand is engaged by hydrophobic interactions with Leu328 and Met448 and parallel π–π stacking with His449 (Fig. 2g). Its tip is near Trp330 and Pro336 on blocking loop 1, while not fully occupying the binding groove. This is in line with compounds featuring sterically more demanding groups instead of the phenyl ring also displaying potent USP30 inhibition[32].

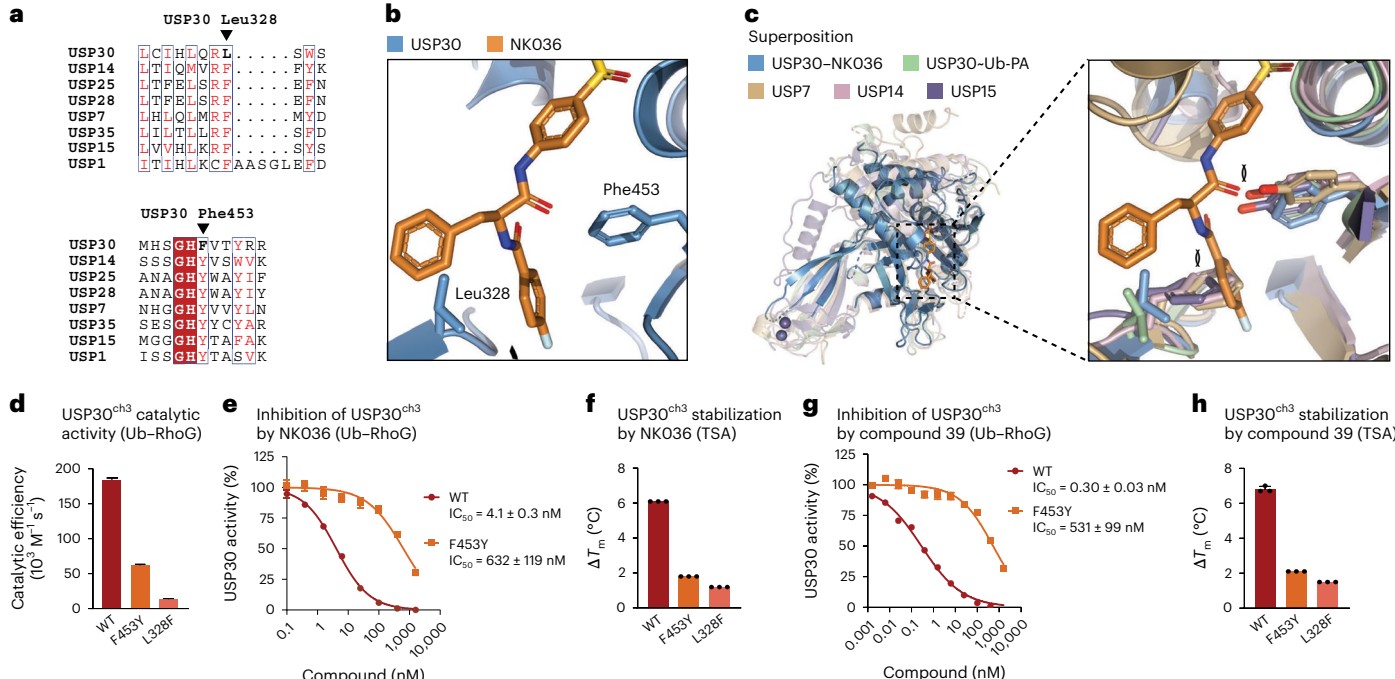

**Fig. 4 | Molecular basis of inhibitor specificity for USP30. a**, Sequence alignment of the indicated human USP DUBs. Arrows indicate the unique Leu328 and Phe453 residues in USP30. **b**, Close-up view of the compound binding site. **c**, Superposition with indicated USP DUB structures in complex with inhibitors (PDB 5N9R, 6IIN, 6GH9), highlighting how equivalent Phe and Tyr residues in other human USP DUBs interfere with compound binding. **d**, Catalytic activities of the indicated wild-type (WT) and mutant USP30 proteins, assessed by Ub–RhoG cleavage. Mean ± s.e.m. (derived from curve fitting; Extended Data Fig. 7). **e**, Inhibitory potencies of NK036, pre-incubated with the indicated USP30 proteins for 1.5 h, determined from Ub–RhoG cleavage assays. $IC_{50}$ values are given as mean ± s.e.m. (derived from curve fitting, with activity data for each concentration recorded as $n = 3$ independent replicates and shown as mean ± s.d.). **f**, Protein stability of the indicated USP30 proteins in the presence of 20 μM NK036. $\Delta T_m$ was calculated as $T_m$ of the compound-bound sample subtracted from $T_m$ of the respective apo protein. Mean ± s.d. ($n = 3$ independent replicates). **g**, Inhibitory potencies of compound 39, determined as in **e**. **h**, Protein stability assessment in the presence of compound 39, determined as in **f**.

3. The benzenesulfonamide is surrounded by Ala162, Phe78, Phe453 and the aliphatic part of the Glu158 side chain (Fig. 2h). The sulfonyl is contacted by the Phe78 and Met79 amides (Fig. 2e). The *tert*-butyl group is bound in a hydrophobic cleft, which is formed toward the top by Met79, Leu165 and Leu144 side chains and on the side by Ile154 and Phe157. The electron density did not allow positioning of the hydroxyl group of NK036 (also reflected in the difference between chain A and chain B geometries; Extended Data Fig. 5b), suggesting that it is not specifically engaged by the protein. This observation in combination with the hydrophobic environment explains the potency (Fig. 1k,l) as well as the tolerance of other hydrophobic structural elements at this position within the compound series[32,34] (Extended Data Fig. 1).

Overall, the structure reveals extensive contacts of all regions of the inhibitor and rationalizes observed structure–activity relationships. The large interaction surface of 570 Å[2] and the deep embedding of the ligand into the protein fold explain the very slow off-rate and previously observed pseudo-covalent binding characteristics[33]. Notably, the regions of the ligand binding site are in perfect agreement with those in the previous HDX-MS analysis. However, the experimentally determined compound binding mode is distinct from docking models obtained from fitting the compound into the Ub-bound geometry of USP30 (ref. 33). This is due to unexpected conformation differences further analyzed below.

## Conformational plasticity of the switching loop and engagement of the Leu73 pocket underlie USP30 ligand binding

To understand how binding to NK036 inhibits USP30, we compared our structure to the USP30-Ub-PA complex (Fig. 3a–d). While the catalytic residues of USP30 are not contacted by the inhibitor, we observed that the catalytic triad was not aligned in the inhibitor-bound state, as the catalytic histidine was flipped out (Fig. 3d). The superposition further showed the phenylalanine and fluorobenzoyl moieties of NK036 to occupy the cleft that guides the ubiquitin C terminus to the USP30 active site. This is accompanied by small changes in blocking loop 2. This substrate competitive binding mode is facilitated by the fluorobenzoyl group binding into a pocket, which is used to recognize the ubiquitin Leu73 side chain, with an inhibitor amide taking the place of the ubiquitin Leu73–Arg74 amide (Fig. 3e,f). NK036 thus inhibits USP30 by preventing ubiquitin binding.

The superposition also revealed large and surprising conformational changes of the switching loop that allow the *N-tert*-butyl-benzenesulfonamide group to bind deeply within the thumb subdomain (Fig. 3d). This cryptic pocket is not present in previously analyzed ubiquitin-bound states. We next wanted to understand how this new conformation is facilitated. Closer inspection of the associated residues revealed that the benzenesulfonamide takes the place of Phe78, which moves inward and takes the position of Ala162. This change in turn pushes the tip of the $\alpha_5$ helix outward by approximately 3 Å to generate an entry into the pocket (Fig. 3g). The switching loop is anchored on the $\alpha_5$ helix by Phe157, which creates a side of the pocket. Notably, Phe157 moves more than 18 Å and takes the place of Trp152, which is pushed out toward the fingers (Fig. 3h and Supplementary Video 1). This surprising conformational change of the switching loop is accompanied by a loop-to-helix transition, with residues 154–159 forming a two-turn α-helix, a conformation not observed for USP30 or any other USP switching loop (Extended Data Fig. 6a–h). As there is no apo structure of USP30 available, we cannot rigorously distinguish between inhibitor-stabilized and de novo inhibitor-induced states.

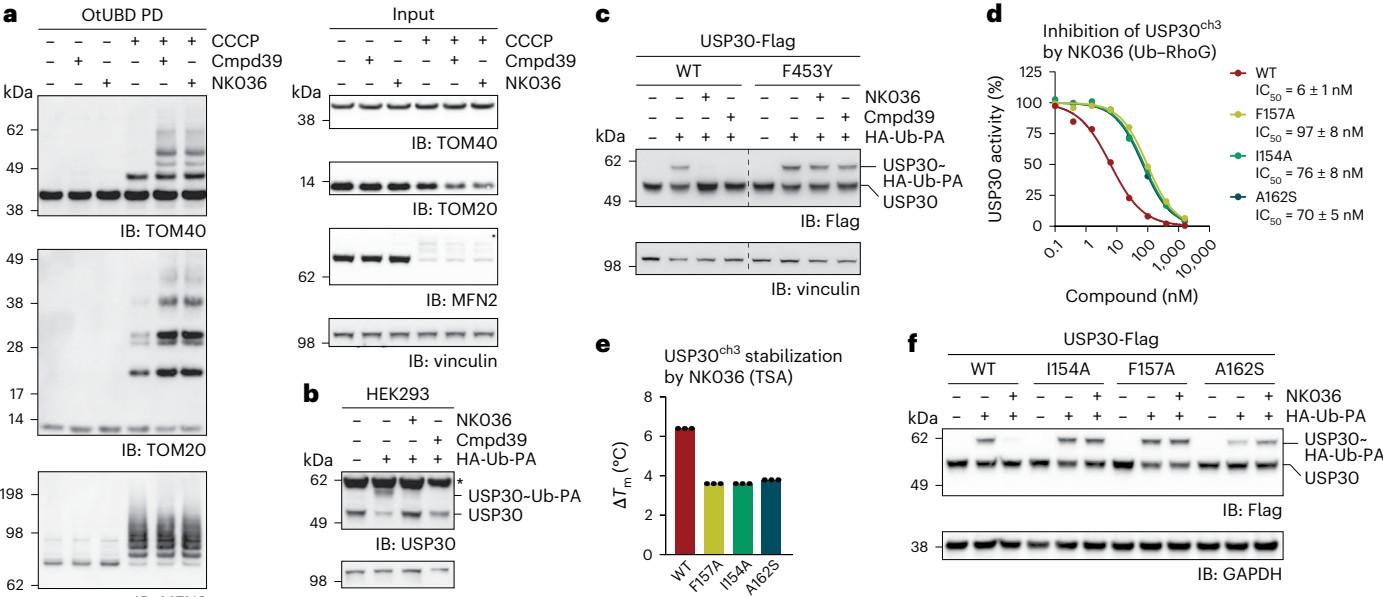

**Fig. 5 | Cellular evaluation of compound-resistant USP30 mutations.**
**a**, Mitochondrial ubiquitination analysis. HeLa cells expressing YFP–parkin were treated with USP30 inhibitors for 19 h where indicated. Mitophagy was induced with carbonyl cyanide *m*-chlorophenyl hydrazone (CCCP) for 1 h. Ubiquitinated proteins were enriched through pulldowns with OtUBD, and samples were analyzed by western blot. Cmpd, compound; IB, immunoblot. **b**, Target engagement assay with endogenous USP30. HEK293 cells were treated with the indicated compounds. Lysates were then incubated with ubiquitin probe where indicated and analyzed by western blot for USP30. The asterisk denotes

an unspecific band. **c**, Cellular assessment of the USP30 inhibition mechanism. C-terminally Flag-tagged USP30 (wild type or with the compound-resistant mutation F453Y) was overexpressed in HEK293 cells. Cells were analyzed as described in **b**, with a western blot for the Flag tag. **d**, Catalytic activities of additional USP30 mutants, assessed by Ub–RhoG cleavage as described for Fig. 4e. **e**, Protein stability of the indicated USP30 proteins by NK036 as described for Fig. 4f. **f**, Cellular probe competition assay as described in **c** with mutations characterized in **d** and **e**. GAPDH, glyceraldehyde-3-phosphate dehydrogenase.

However, the observations that the switching loop shows very high flexibility by HDX-MS in the apo state and that the observed conformation is incompatible with ubiquitin binding strongly suggest that this is an inhibitor-induced conformation, which is facilitated by structural plasticity of the switching loop. This conformational change also represents a second mechanism by which ubiquitin engagement is prevented in the inhibitor-bound state.

**Molecular basis for specific inhibition of USP30**
We next asked why the inhibitor series displays pronounced specificity for USP30 both in vitro and in cells[27,33]. Analysis of the binding site revealed that many residues contacted by the inhibitor are strictly conserved in many other USP family members[39]. These include Phe78 and Met79 directly adjacent the catalytic cysteine, His444 on blocking loop 2, Ala162 at the start of $\alpha_5$ and Tyr495. However, two residues stood out that are unique for USP30 within the entire human USP family: Leu328 where other USP DUBs typically feature a phenylalanine and Phe453 instead of the common tyrosine (Fig. 4a). Both residues were noted previously during the analysis of atypical ubiquitin binding by USP30 (ref. 37) and are located within the center of the ligand binding site (Fig. 4b). Superpositions with structures of other DUBs featuring the canonical USP residues at these positions indicate how these would interfere with ligand binding (Fig. 4c). The additional phenolic hydroxyl group of a tyrosine at position 453 would clash with the carbonyl oxygen of a ligand amide, whereas a larger phenylalanine at position 328 would interfere with two of the three hydrophobic pockets due to Leu328's role as a pin between these. This analysis suggested that Leu328 and Phe453 are key specificity factors for USP30 inhibition by benzenesulfonamide inhibitors.

To experimentally test this hypothesis, we generated USP30^F453Y and USP30^L328F point-mutated proteins. While the reduced catalytic activity of USP30^L328F did not allow enzyme inhibition assays, the

USP30^F453Y mutant showed reduced but robust cleavage of Ub–RhoG (Fig. 4d and Extended Data Fig. 7a,b). Substitution of Phe453 by the canonical tyrosine reduced inhibition potency of NK036 by more than two orders of magnitude (Fig. 4e), which validates the observed binding mode. Consistently, both mutant proteins were less stabilized by NK036 in thermal shift assays. Thermal shift assays were carried out at a compound concentration (20 μM) above the inhibitory IC$_{50}$, explaining partial protein stabilization (Fig. 4f). We also repeated the analysis with compound 39 with identical results, identifying compound-resistant activity of the USP30^F453Y mutant with a specificity window of three orders of magnitude (Fig. 4g,h). These data demonstrate that two residues that are unique for USP30 within the human USP DUB family and that are in the center of the inhibitor binding site are key to facilitating specific inhibition of USP30.

We next set out to assess this mechanism of USP30 inhibition in a cellular context. To this end, we first determined that NK036 acts on endogenous USP30. We used the established mitophagy model system of HeLa cells constitutively expressing yellow fluorescent protein (YFP)–parkin[5] and applied the recently introduced ubiquitin binding domain derived from an *Orientia tsutsugamushi* DUB (OtUBD) for enrichment of ubiquitinated proteins[52,53]. We observed increased ubiquitination of USP30 substrates upon USP30 inhibition shortly after mitochondrial depolarization[11,13,16,22,37] (Fig. 5a). Specifically, compound 39 and NK036 elevated the polyubiquitination of translocase of the outer membrane subunits 20 and 40 (TOM20 and TOM40), while not affecting the polyubiquitination of mitofusin 2 (MFN2). The activity of both compounds was also confirmed through a probe-labeling experiment with endogenous USP30 (ref. 27) (Fig. 5b). Next, we optimized a competitive probe-labeling assay. Pre-incubation of inhibitors with wild-type recombinant USP30 largely abrogated labeling with Ub-PA, whereas USP30 carrying the compound-resistant F453Y mutation showed complete labeling in all conditions (Extended Data Fig. 7c).

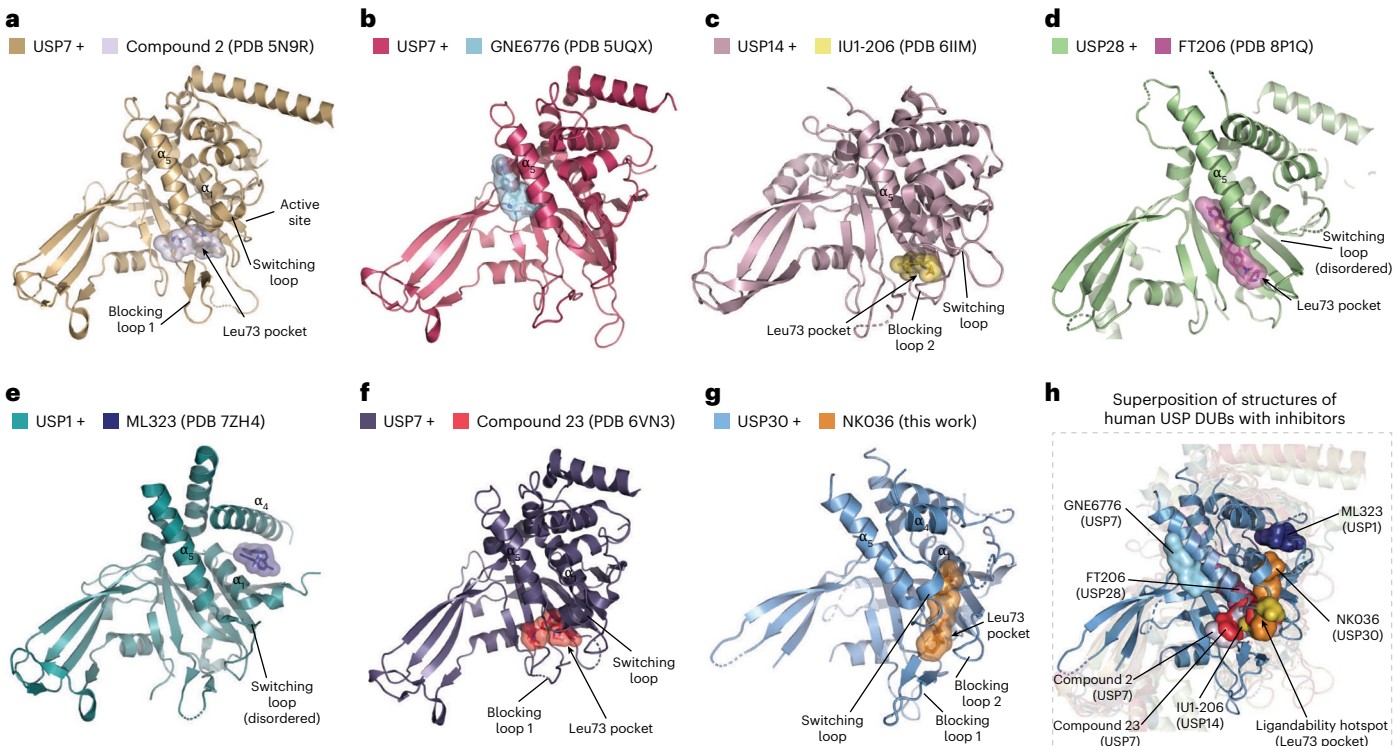

**Fig. 6 | Inhibition of USP30 by NK036 uses a ligandability hotspot and a cryptic pocket, distinct from other USP DUB inhibitors. a–g**, Cartoon representations of human USP family DUB catalytic domains in complex with the indicated small-molecule inhibitors (USP7 and compound 2 (PDB 5N9R; **a**), USP7 and GNE6776 (PDB 5UQX; **b**), USP14 and IU1-206 (PDB 6IIM; **c**), USP28 and FT206 (PDB 8P1Q; **d**), USP1 and ML323 (PDB 7ZH4; **e**), USP7 and compound 23 (PDB 6VN3; **f**), USP30 and NK036 (**g**)). Compounds are shown as both sticks and transparent surfaces. Structural elements of DUBs are labeled, and PDB codes of structures are given[44,45,48,54,56,57]. The Leu73 ubiquitin binding site is shown with an arrow when engaged by compounds. **h**, Comparison of USP30 inhibition by NK036 to other DUB inhibitors. Superposition of the structure of USP30 + NK036 on other structures shown in **a–f**. Compounds are shown as surfaces and are labeled. All USP cartoons except USP30 are semitransparent.

We then overexpressed Flag-tagged, full-length USP30 in HEK293 cells to compare probe labeling of wild-type and mutant USP30 (Fig. 5c). Treatment of cells for 2 h with either compound 39 or NK036 abolished probe labeling of the wild-type protein. By contrast, USP30 carrying the F453Y mutation was equally labeled by the ubiquitin probe when cells were previously treated with inhibitors. To assess the importance of the USP30 switching loop and the cryptic pocket, we selected three additional compound-resistant mutations based on Phe157 (its large conformation change is described in Fig. 3), Ala162 (the gate-keeper residue of the cryptic pocket) and Ile154 (in the upper part of the thumb domain, contacting the sulfonamide) (Fig. 3 and Extended Data Fig. 7d). In vitro assays confirmed reduced inhibitory potency as well as reduced stabilization by compounds for the USP30[F157A], USP30[I154A] and USP30[A162S] mutations (Fig. 5d,e and Extended Data Fig. 7e,f). Consistently, we also observed in cells that overexpressed USP30 carrying these mutations was refractory to inhibition by NK036 (Fig. 5f and Extended Data Fig. 7g). In sum, these data confirm the compound resistance of these mutations in the context of the full-length protein. They provide cellular validation for the observed mechanism of specific USP30 inhibition by benzenesulfonamides.

## NK036 engages the DUB ligandability hotspot in a unique manner

Two structures of USP30 in complex with covalent inhibitors are available through the PDB. These were determined with a previously reported construct, which was stabilized through a custom antibody fragment. However, due to the lack of an associated manuscript, accessory information, for example, on the antibody, validation of the binding mode or compound specificity, is lacking. We carried out a comparison of the binding modes of covalent and non-covalent USP30 inhibitors (Extended Data Fig. 8a–h). Importantly, both binding modes are drastically different, with unique conformational changes of the switching loop associated with both. This comparison shows that both binding modes are very distinct, and the obtained data are thus highly complementary.

To understand how the identified mechanism of specific USP30 inhibition differs from those of other non-covalent USP DUB inhibitors, we analyzed published human USP–ligand structures. These included USP7 in complex with hydroxypiperidine inhibitors[46–48] (Fig. 6a), USP7 in complex with an allosteric inhibitor[45] (Fig. 6b), USP14 bound to IU1-206 (ref. 44; Fig. 6c), USP28 in complex with FT206 (refs. 54,55; Fig. 6d), USP1 in complex with ML323 (ref. 56; Fig. 6e) and USP7 in complex with compound 23 (ref. 57; Fig. 6f). In addition, we also compared the binding mode to a covalent USP7 inhibitor[46] as well as to an inhibitor of the USP fold severe acute respiratory syndrome coronavirus (SARS-CoV) papain-like protease (PLpro) enzyme[58,59] (Extended Data Fig. 9a–d). Superposition of all structures with the obtained USP30–NK036 geometry revealed that the benzenesulfonamide moiety of NK036 engages a previously unexplored pocket within the USP domain (Fig. 6g,h).

The fluorophenyl and phenylalanine moieties of NK036 overlap with binding modes determined for non-covalent USP7, USP14 and USP28 inhibitors as well as for covalent USP7 and PLpro inhibitors. Closer analysis of the structural superpositions revealed that the *para*-chlorophenyl-fluorophenyl group of NK036 engages the Leu73 ubiquitin binding site of USP30 in the same way as the chemically related *para*-chlorophenyl-fluoropyrazole and 3-fluoropyrazole groups of USP7 and USP14 inhibitors, which are otherwise structurally completely unrelated[44,46,57] (Extended Data Fig. 10). We previously identified the molecular basis for specific inhibition of the UCH family DUB UCHL1 by covalent cyanamides[60,61] and, in this context, proposed the

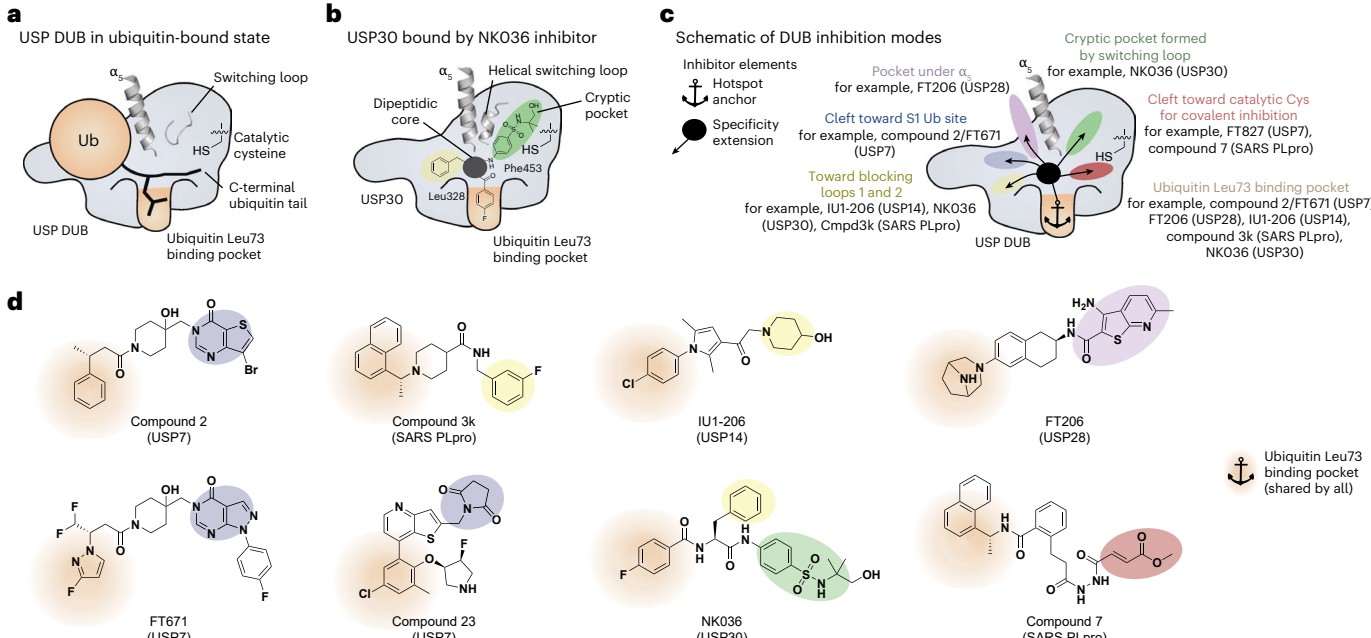

**Fig. 7 | A framework for specific DUB inhibition. a**, Schematic of a ubiquitin-bound USP DUB. Structural elements are labeled. **b**, Schematic of USP30 in complex with an inhibitor of the benzenesulfonamide scaffold, occupying the ubiquitin Leu73 pocket (sand), the cryptic pocket (green) and the cleft toward the S1 ubiquitin binding site (blue) with shown chemical moieties. **c**, Schematic of DUB inhibition with compounds being composed of a hotspot anchor element (occupying the ubiquitin Leu73 pocket) and one or two specificity extensions (occupying the other shown USP DUB structural elements). Compounds featuring the respective extensions are named together with their cognate DUBs[44,46,48,54,55,57,58]. See Extended Data Fig. 10 for structural superpositions focused on the Leu73 binding pocket. **d**, Chemical structures of specific DUB inhibitors. Chemical motifs occupying the distinct binding sites are colored according to **c**. The hotspot anchor motifs shared by all are highlighted.

binding pocket of DUBs for the ubiquitin Leu73 side chain as a general DUB ligandability hotspot[60]. Our analysis reveals that NK036 engages USP30 through precisely this hotspot in a unique manner as it also expands through its benzenesulfonamide moiety into a cryptic pocket in the thumb subdomain, which is generated through conformational plasticity of the switching loop.

## Discussion

Due to the multifaceted cellular roles of the ubiquitin system, many novel therapeutic approaches are currently being explored by modulating ubiquitin signaling with small molecules[62]. DUB inhibitors have the potential to amplify ubiquitin-dependent processes, and inhibitors of two DUBs are currently being evaluated in clinical trials[24,63]. These include compounds targeting the mitochondrial DUB USP30 for kidney disease and PD to elevate mitochondrial quality control, yet how the specific inhibition of USP30 can be facilitated on the molecular level had remained elusive. Through chimeric engineering, we here report a structure of human USP30 in complex with a potent inhibitor. The devised constructs will also aid the structural investigation of other USP30 inhibitors, and the obtained structure will enable rational design and discovery of improved compounds while retaining the excellent potency and specificity of the scaffold.

Our investigation reveals the molecular basis for potent and specific inhibition of USP30. NK036 uses its fluorobenzoyl moiety to occupy the unique Leu73 recognition pocket of USP30 and additionally occupies a cryptic pocket within the thumb subdomain with its benzenesulfonamide group (Fig. 7a,b). This binding mode is facilitated by a loop-to-helix transition of the switching loop, for which solvent-exposed residues are repurposed to lock the loop in a structured, domain-bound state. This loop shows high sequence variability between human USP DUBs and has previously been discussed in the context of specific USP7 inhibition, where compounds of the hydroxypiperidine scaffold stabilize the inactive apo state of USP7 (refs. 46,47).

Distinctly, inhibition of USP30 by benzenesulfonamides is facilitated by a new inhibitor-induced conformation of the switching loop (Fig. 3 and Extended Data Fig. 8). This sets the binding mode identified here apart from all other DUB–ligand complexes. While inhibitor-bound states of UCHL1 (refs. 60,61) appear as hybrids of apo- and ubiquitin-bound states, the new NK036-engaged conformation prevents ubiquitin engagement in addition to facilitating compound binding. This mechanism is also distinct from the displacement of a loop in USP1 by ML323 (ref. 56). Our study highlights that conformational changes of the DUB need to be considered when assessing modes of inhibition and thus complements a recent classification of DUB inhibitor binding sites[41].

It is striking how the hydrophobic pocket for the ubiquitin Leu73 side chain has emerged as a central ligandability hotspot across DUB families to anchor chemically diverse inhibitors, which then extend from there into different areas of the catalytic domains to achieve selective DUB inhibition (Figs. 6h and 7, and Extended Data Figs. 9 and 10): covalent USP7, PLpro and UCHL1 inhibitors extend toward the catalytic cysteines to bind these with electrophilic handles, whereas non-covalent USP7, USP14 and PLpro inhibitors extend into the opposite direction toward the S1 ubiquitin binding site and the blocking loops. USP28 inhibitors extend into a pocket underneath the α5 helix, while NK036 extends into a cryptic pocket formed by the switching loop adjacent to α5.

Based on this analysis, we propose a conceptual framework for specific USP DUB inhibition that is centered on the engagement of a common ligandability hotspot in the Leu73 ubiquitin binding site paired with diverse compound extensions (Fig. 7c,d). Importantly, the benzenesulfonamide scaffold stands out among DUB inhibitors with its tripartite geometry. It demonstrates that the simultaneous presence of two such compound extensions is possible, which explains the large interaction area and high potency. The resulting distinct binding modes explain compound specificities and, with more structures

emerging, may enable rational scaffold-hopping approaches to design novel DUB inhibitors.

Excitingly, this analysis also allowed the identification of highly related chemical moieties that occupy hydrophobic Leu73 ligandability hotspots in structurally otherwise unrelated DUB inhibitors for different enzymes (compare the *para*-chlorophenyl groups in the USP7 and USP14 inhibitors compound 23 and IU1-206, respectively, and the 4-fluorophenyl group in NK036 as hotspot anchor elements; Fig. 7d and Extended Data Fig. 10b,c). This chemical feature thus appears to be a privileged structure for non-covalent DUB inhibition. Our analysis and the discovered geometry thus establish a roadmap toward specific inhibitors also for other DUBs, contributing to an emerging logic of DUB inhibition akin to more established inhibitor classifications for, for example, kinases. The shared engagement of the Leu73 binding pocket in DUBs reflects the central role of Leu73 recognition for DUB activity, which is evident also from the DUB-resistant L73P mutation in ubiquitin[64].

We here also introduce a generalizable chimeric engineering strategy to increase the crystallization propensity of USP catalytic domains. This approach expands previously explored strategies for DUB crystallization enhancement including insertion deletion for USP4, USP25, USP28 and USP30 (refs. 37,54,55,65,66), domain swapping for USP11 (ref. 67), and surface residue mutations for USP9X and USP30 (refs. 37,68). It also synergizes with recent reports on the engineering of the cereblon E3 ligase[69,70] for streamlined structural characterization. We expect that our approach will form a platform to enable structure-based drug design more broadly for USP DUBs and possibly other ubiquitin-dependent enzymes.

Collectively, our work illuminates the potential of cryptic pockets and conformational dynamics to obtain specific DUB inhibitors and advances the structure-based design of therapeutics targeting neurodegenerative diseases.

## Online content

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

## Methods

### Cloning and protein expression

Chimeric USP30 constructs for bacterial expression were generated by amplifying protein-coding parts from plasmids or by incorporating sequences as overhangs into primers. The following sequences were used: codon-optimized human USP30 catalytic domain, Addgene 110746, UniProt Q70CQ3 with modifications described previously[37] and in Supplementary Information; human USP7, UniProt Q93009; human USP14, UniProt P54578; human USP35, UniProt Q9P2H5; human CYLD, UniProt Q9NQC7. All protein sequences are shown in Supplementary Information. DNA fragments were joined through SOE-PCR and subsequently ligated into the pOPINK vector using the In-Fusion HD Cloning Kit (Takara Clontech). A pOPINE vector containing sequence for C-terminally Flag-tagged human USP30 (1–517) was used for transient transfection as described previously[37]. Point mutants were introduced through vector QuikChange or SOE-PCR. Constructs were confirmed by Sanger sequencing.

Bacterial expression was performed in Rosetta 2(DE3)pLacI cells. An overnight culture was diluted 1:100 in 2xTY medium supplemented with appropriate antibiotics. Cultures were grown at 30 °C with shaking at 180 rpm until an optical density ($OD_{600}$) of 0.8–1.0 was reached. After cooling to 18 °C, protein expression was induced by adding isopropyl β-D-thiogalactopyranoside at a final concentration of 0.5 mM, and cells were kept shaking at 18 °C for 18 h. Collected bacterial cells were stored at −80 °C.

### Protein purification

Cells were resuspended in lysis buffer (50 mM sodium phosphate, pH 8.0, 300 mM NaCl, 20 mM imidazole, 4 mM β-mercaptoethanol) supplemented with DNase I and lysozyme. The cell suspension was lysed by sonication (55% amplitude, 10 s on and 10 s off) on ice, cleared by centrifugation at 33,500g for 30 min at 4 °C and filtered through a 0.45-μm filter.

All purification steps were performed on ÄKTA pure systems (GE Healthcare) at 4 °C. For affinity chromatography, lysate was loaded onto a pre-equilibrated 5-ml HisTrap fast flow column (GE Healthcare), washed with 20 column volumes of lysis buffer and then eluted with a linear gradient of 3.5 column volumes into elution buffer (50 mM sodium phosphate, pH 8.0, 300 mM NaCl, 500 mM imidazole, 4 mM β-mercaptoethanol). Protein-containing fractions were pooled, supplemented with His$_6$-tagged 3C protease and dialyzed against a solution of 25 mM Tris, pH 8.5, 100 mM NaCl and 4 mM dithiothreitol (DTT) overnight at 4 °C. The dialyzed sample was diluted twofold with 20 mM Tris, pH 8.5, filtered through a 0.45-μm filter and directly loaded onto a 6-ml Resource Q column, equilibrated with 25 mM Tris, pH 8.5, 50 mM NaCl, 4 mM DTT, for anion exchange chromatography. Elution was achieved with a gradient into high-salt buffer (25 mM Tris, pH 8.5, 500 mM NaCl, 4 mM DTT). Further purification was carried out on a HiLoad 16/600 Superdex 75 pg column in 20 mM Tris, pH 8.0, 100 mM NaCl, 4 mM DTT for samples used for crystallography. Purity of peak fractions was assessed by SDS–PAGE before pooling, concentrating at 4 °C and 3,200g in spin concentrators (10-kDa MWCO, Amicon) and flash freezing in liquid nitrogen. Protein concentrations were determined on a NanoDrop (Thermo Fisher).

### Construct design and modeling

Protein construct design was guided by the annotation of USP domain boxes[39] and available crystal structures of human USP DUBs including those of USP30 (refs. 37,38), USP7 (refs. 42,45–47), USP14 (refs. 43,44), USP35 (ref. 51) and CYLD[49] (Supplementary Table 1). Chimeric USP30 protein sequences were based on the sequence used for PDB entry 5OHK (ref. 37). Structures of chimeric sequences were predicted with AlphaFold2 (ref. 71) and analyzed with PyMOL. Proteins were then experimentally prepared and biochemically characterized in an iterative process.

### Crystallization

Protein samples for crystallization were prepared by mixing USP30 (30 mg ml⁻¹ in gel filtration buffer) with 1.4 equivalents of NK036, which was added in 2 steps from a 100 mM DMSO stock. Following incubation for 30 min at room temperature, the sample was passed through a Proteus Mini Clarification Spin Column (Protein Ark) and used directly for crystallization trials. Screening plates were set up by a mosquito HTS robot (TTP Labtech) in 96-well sitting-drop vapor diffusion plates in MRC format (Molecular Dimensions) and incubated at 20 °C. Drop ratios of 200 nl + 200 nl and 500 nl + 500 nl were used for coarse screening and fine screening, respectively.

Initial crystals of USP30 construct ch3 in complex with NK036 were found in 0.1 M bicine, pH 9.0, 5% (wt/vol) PEG 20,000, 1% (vol/vol) dioxane. Moderately sized cylindrical crystals (60 × 16 × 16 μm³) were obtained through fine screening in 88 mM NaOH, 100 mM bicine, 11% (wt/vol) PEG 20,000, 1% (vol/vol) dioxane.

Crystal size and diffraction quality were further improved through the Hampton Additive Screen on the above-mentioned condition. Larger cylindrical crystals (130 × 25 × 25 μm³) of diffraction quality were obtained in 76 mM NaOH, 100 mM bicine, 10.2% (wt/vol) PEG 20,000, 1% (vol/vol) dioxane, 10 mM L-proline (crystal 1) and 85 mM NaOH, 100 mM bicine, 10.2% (wt/vol) PEG 20,000, 1% (vol/vol) dioxane, 10 mM L-proline (crystal 2) and 88 mM NaOH, 100 mM bicine, 11% (wt/vol) PEG 20,000, 1% (vol/vol) dioxane, 10 mM sarcosine (crystal 3). Crystals were soaked in mother liquor supplemented with 25% (vol/vol) glycerol (crystals 1 and 2) or ethylene glycol (crystal 3) before vitrification in liquid nitrogen.

The covalent ubiquitin complex was obtained by mixing USP30 after ResQ purification with 1.2 equivalents of Ub-PA. Following incubation at room temperature for 2 h, USP30-Ub-PA was purified on a HiLoad 16/600 Superdex 75 pg column as described above and concentrated to 12 mg ml⁻¹. Diffraction-quality crystals were obtained from coarse-screening plates in 0.56 M sodium citrate, pH 7.0. Crystals were soaked in mother liquor supplemented with 25% (vol/vol) ethylene glycol before vitrification in liquid nitrogen.

### Data collection, structure solution and refinement

Diffraction data were collected at 100 K at the ESRF on beamline ID30A-3. Datasets from different crystals of the USP30–NK036 complex were separately integrated with DIALS[72]. Multicrystal data averaging was employed as implemented in CCP4 BLEND[73] to improve resolution, lower B factors and ultimately enhance features in the electron density. Multiple datasets and combinations were explored, resulting in the chosen data averaged from crystals 1–3 as the best cluster. The final combined data were then corrected for anisotropy by the STARANISO webserver[74]. Table 1 displays data collection and refinement statistics, and data for anisotropically scaled individual datasets are provided in Supplementary Table 3 for comparison. The final structure was solved to a resolution of 2.75 Å from the averaged data by molecular replacement using MR Phaser[75] and a search model based on the AlphaFold2-derived structure model of the chimera. The final model was obtained by multiple cycles of structure building in Coot[76] and refinement with Phenix Refine[77]. The compound geometry was optimized with ORCA[78] at the B3LYP/def2-SVP level of theory. Geometry restraints were generated by eLBOW[79].

Diffraction data leading to the USP30-Ub-PA structure were integrated with DIALS. The final structure was obtained by molecular replacement with MR Phaser and refinement with Phenix Refine as described above. Final data collection and refinement statistics are shown in Table 1.

### Ub-PA labeling assay

Purified proteins were diluted in 20 mM Tris, pH 8.0, 300 mM NaCl, 2 mM DTT, 5% glycerol. Protein and probe were combined at final concentrations of 2.5 μM and 10 μM, respectively, and reacted for 1 h at

room temperature. For the in vitro compound–probe competition assay, 1 μM protein was pre-incubated with 8 μM compound at room temperature for 2 h. Following compound incubation, 4 μM probe was added and reacted for 1 h at 37 °C. Probe binding was assessed by SDS–PAGE and Coomassie staining.

## Ubiquitin–RhoG cleavage assay
Enzyme kinetics were performed in black, low-volume, non-binding-surface 384-well plates. For activity assays, 2× Ub–RhoG substrate (final concentration, 50 nM) and 2× enzyme (final concentration, 0.25–32 nM) were prepared in 20 mM HEPES, pH 8.0, 50 mM NaCl, 5 mM DTT, 0.1 mg ml$^{-1}$ BSA. Next, 10 μl of 2× enzyme solution was added in triplicate to the plate. The reaction was started by adding 10 μl of 2× substrate solution. For enzyme inhibition assays, 2× substrate (final concentration, 100 nM), 4× enzyme (final concentration, 0.1 nM) and 4× inhibitor (final concentration, 1.6 pM–1.6 μM) were prepared in 50 mM Tris-HCl, pH 8.0, 0.05 mg ml$^{-1}$ BSA, 4 mM DTT, 0.01% Tween-20. Inhibitor solution (4×) was added to 4× enzyme solution to create a 2× enzyme–inhibitor mixture. Next, 10 μl of this 2× enzyme–inhibitor mixture was then added to the plate in triplicate, and the plate was incubated at room temperature for 90 min. The reaction was started by adding 10 μl of 2× Ub–RhoG substrate. Substrate cleavage was monitored by measurement of cleaved RhoG fluorescence (excitation, 492 nm; emission, 525 nm) every 30 s for 1 h at 25 °C on a Tecan Spark plate reader. Data were analyzed with GraphPad Prism.

## Thermal shift assay
The thermal stability of USP30 constructs and effects of either inhibitor or Ub-PA binding were determined by thermal shift assays. Inhibitor, Ub-PA and SYPRO orange dye (Sigma, 5,000× stock in DMSO) solutions were prepared in 1× PBS, 4 mM DTT. Inhibitor/Ub-PA was prediluted to 100 μM, and SYPRO orange dye was prediluted to 50× (from the commercial stock). Mixtures of protein, inhibitor/Ub-PA and SYPRO were prepared with inhibitor/Ub-PA at a final concentration of 20 μM, SYPRO at a final concentration of 4× and protein at a final concentration of 2 μM for c1, 4 μM for ch1, ch4 and c2 and 3 μM for ch2 and ch3. Samples were added in triplicate to a white 96-well PCR plate (Bio-Rad). Fluorescence intensity was monitored (excitation, 450–490 nm; emission, 560–580 nm) at a gradient of 20–90 °C (increment of 0.3 °C, hold for 5 s before read) on a real-time PCR system (Bio-Rad).

## Cell culture
HEK293 cells (ACC 305) were obtained from the Leibniz Institute DSMZ German Collection of Microorganisms and Cell Cultures. HeLa cells constitutively expressing YFP–parkin were kindly provided by R. Youle (National Institute of Neurological Disorders and Stroke, Bethesda, Maryland). Cells were cultured in DMEM medium with high glucose supplemented with 10% FBS and penicillin–streptomycin. Cells were grown at 37 °C in a humidified atmosphere with 5% (vol/vol) CO$_2$. Cells were tested to be free of *Mycoplasma* contamination.

## Transfection
HEK293 cells ($8 \times 10^5$ cells per well) were seeded in six-well plates and cultivated for 24 h. PEI transfecting reagent (Polysciences) was pre-incubated for 10 min with the vectors in 200 μl Opti-MEM medium (2 μg DNA per well). Cells were transfected and incubated for 48 h. Following the treatment with either compounds or DMSO in fresh medium for an additional 2 h, cells were processed as described below.

## Cellular ubiquitin probe competition
Cells were seeded at $8 \times 10^5$ cells per well and grown for 48 h. Compounds (2 μM for overexpressed USP30, 8 μM for endogenous USP30) or DMSO were added for 2 h in freshly supplied medium. The medium was aspirated, and cells were washed once with ice-cold PBS. Cells were collected by scraping in lysis buffer (150 mM NaCl, 250 mM sucrose,

50 mM Tris, pH 8.0, 5 mM TCEP) supplemented with EDTA-free complete protease inhibitor cocktail and benzonase. Suspended cells were homogenized by 12 bursts of a microtip probe sonicator (Branson Sonifier 450) at 10% amplitude with pulses of 1 s on ice. Remaining cell debris was separated by centrifugation (20,817$g$, 5 min, 4 °C). HA-Ub-PA probe was added to the cleared lysate (approximately 2–4 mg ml$^{-1}$) at a final concentration of 10 μM when indicated. After incubation for 10 min at room temperature, reactions were quenched by the addition of 4× reducing LDS sample buffer. Samples were resolved by SDS–PAGE and further analyzed via western blot.

## Western blotting
Using a Trans-Blot Turbo system (Bio-Rad, 1.0 A, 25 V, 30 min), proteins were blotted onto a nitrocellulose membrane. Membranes were saturated by incubating in 5% (wt/vol) non-fat milk in PBS-T. Primary antibodies (anti-Flag, 1:1,000, Sigma, F3165; anti-USP30, 1:500, Sigma, HPA016952; anti-vinculin, 1:10,000, Sigma, V9131; anti-GAPDH, 1:10,000, Thermo Fisher, AM4300; anti-TOM40, 1:2,000, Proteintech, 18409-1-AP; anti-TOM20, 1:5,000, Proteintech, 11802-1-AP; anti-MFN2, 1:5,000, Proteintech, 12186-1-AP) were allowed to bind overnight. Appropriate horseradish peroxidase-coupled secondary antibodies (anti-mouse, 1:5,000, Sigma, NXA931; anti-rabbit, 1:5,000, Sigma, GENA934) were applied, and chemiluminescence was generated using Clarity Western ECL Substrate (Bio-Rad). For detection of endogenous USP30 levels, the Clarity Western ECL Substrate was supplemented with 10% Clarity Max Western ECL Substrate (Bio-Rad). Images were recorded on a ChemiDoc MP Imaging System (Bio-Rad).

## Enrichment of ubiquitinated proteins
HeLa cells constitutively expressing YFP–parkin were treated with 2 μM of the respective compounds or an equivalent volume of DMSO on the day after seeding. Nineteen hours after compound treatment, 10 μM carbonyl cyanide *m*-chlorophenyl hydrazone (CCCP, Acros Organics) or DMSO of the equivalent volume was added for 1 h before cells were washed with ice-cold PBS. Cells were then lysed in urea lysis buffer (4 M urea, 50 mM Tris-HCl, pH 8, 150 mM NaCl, 1% (vol/vol) IGEPAL, 2 mM EDTA, 5% (vol/vol) glycerol, 1× EDTA-free protease inhibitor cocktail, 1 mM PMSF, 10 μM PR619, 20 mM *N*-ethylmaleimide, 4 mM 1,10-phenanthroline) and collected by scraping. Cells were homogenized by sonication for 10 s (2 s on and 2 s off) at 10% amplitude and centrifuged for 10 min at 14,000$g$ and 4 °C.

Biotinylated OtUBD (2 nmol, prepared as described elsewhere[53]) was immobilized on 35 μl of a high-capacity NeutrAvidin agarose bead slurry (Pierce, Thermo Fisher Scientific) per condition for 1 h at 4 °C with rotation. Excess reagent was washed away with ice-cold PBS. The beads were then equilibrated in urea lysis buffer. For each condition, an equal amount of 3 mg protein (as determined by the Bradford assay, in 1 ml of lysis buffer, typically obtained from one 10-cm dish) was added to the beads. These were then incubated for 2 h at 4 °C with rotation and pelleted by centrifugation at 500$g$ for 1 min, and the supernatant was removed. Beads were then washed once with lysis buffer, once with high-salt buffer (50 mM Tris-HCl, pH 8, 1 M NaCl) and twice with PBS. Elution was carried out using 2× LDS sample buffer with 25 mM DTT and boiling for 5 min at 95 °C. After pelleting the beads at 500$g$ for 1 min, the supernatant was transferred and analyzed by SDS–PAGE and western blotting as described above.

## Chemical synthesis
Procedures for the synthesis of the inhibitors compound 39 (ref. 27) and NK036 as well as compound characterization data are reported in Supplementary Information. The synthesis was based on previously described methods[32] for the synthesis of compound 39, with adjustments based on the synthesis of a related compound[80]. A different synthesis route of the title compound NK036 (termed I-137) was published elsewhere[34].

## Statistics and reproducibility

All observations reported in this study are based on at least two, typically three or more biologically independent experiments with consistent results.

## Reporting summary

Further information on research design is available in the Nature Portfolio Reporting Summary linked to this article.

## Data availability

Coordinates and structure factors for the USP30ch3 + NK036 and USP30ch3-Ub-PA crystal structures were deposited in the PDB under accession codes 9F19 and 9F6G, respectively. Coordinates of other structures were obtained from the PDB through accession codes 2AYN, 5N9R, 5OHK, 5OHP, 5TXK, 5UQX, 6GH9, 6IIM, 6IIN, 6VN3, 7ZH4 and 8P1Q. Source data are provided with this paper.

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

## Acknowledgements

We acknowledge beamtime at the ESRF and thank R. Gasper and P. Geue for support with crystallization and biophysics. We are grateful to the Max Planck Computing and Data Facility for the provision of computing resources. We thank all staff at the MPI Dortmund and at TU Dortmund University for excellent support. We are grateful to all members of the Gersch laboratory for discussions, advice and reagents. We thank R. Youle (NIH Bethesda) for providing HeLa cells expressing YFP–parkin. Gifts of plasmids from D. Komander (MRC LMB Cambridge, WEHI Melbourne), G. Maertens (Imperial College London) and G. Peters (CRUK London) are gratefully acknowledged. This work was funded by the Chemical Genomics Centre (AstraZeneca, Merck, Pfizer and the Max Planck Society, CGCIII-352S to M.G.) and the German Research Foundation through CRC1430 (DFG, 424228829 to M.G.). Work in the Gersch laboratory is further supported by an Emmy Noether grant from the German Research Foundation (DFG, GE 3110/1-1 to M.G.) and by the State of North Rhine-Westphalia through the 'CANcer TARgeting Network' (NW21-062C to M.G.). Any syntheses or derivatizations of patented compounds were used for research purposes only. The funders had no role in study design, data collection and analysis, decision to publish or preparation of the manuscript.

## Author contributions

N.H.K. performed all experiments unless noted otherwise. N.K. synthesized chemical compounds. K.G. and G.-M.K. performed cellular experiments. All authors designed experiments, analyzed data, interpreted results and generated figures. M.G. conceived of the study, supervised research and wrote the paper with input from all authors.

## Funding

## Competing interests

The authors declare no competing interests.

## Additional information

**Extended data** is available for this paper at https://doi.org/10.1038/s41594-025-01534-4.

**Correspondence and requests for materials** should be addressed to Malte Gersch.

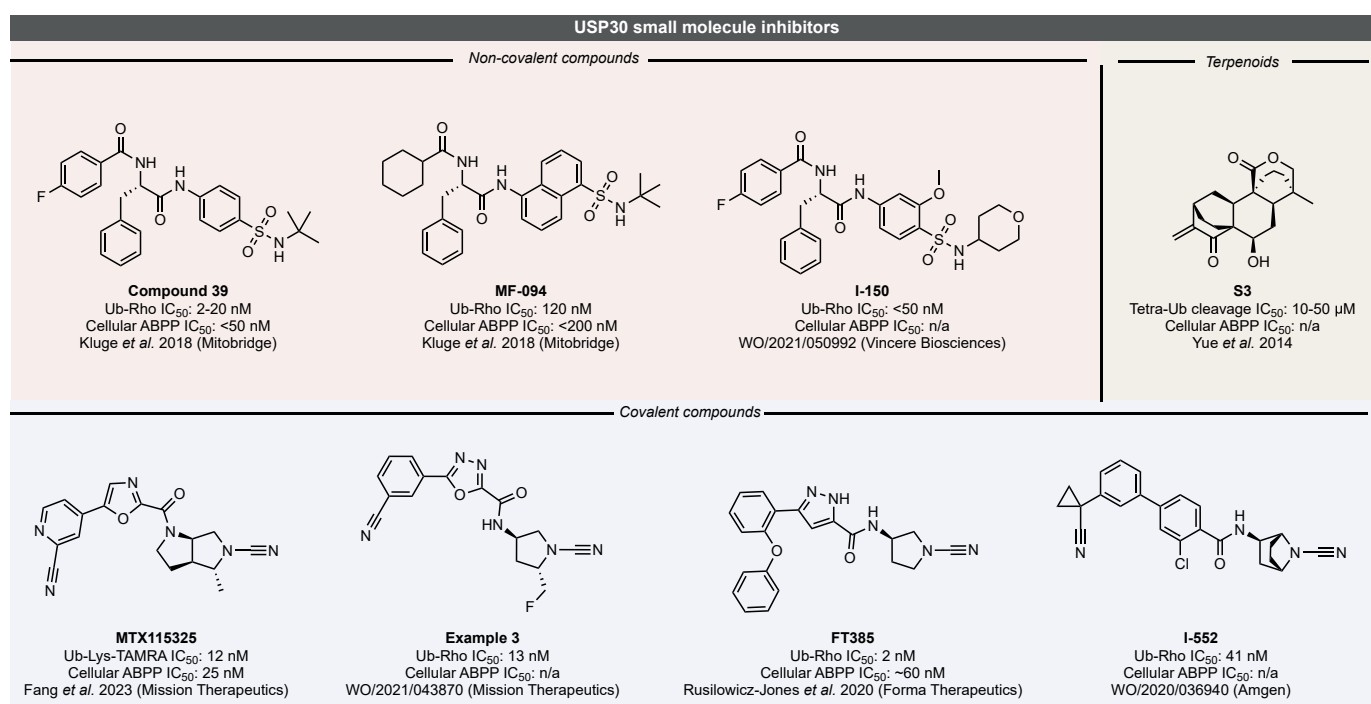

**Extended Data Fig. 1 | Small molecule inhibitors of USP30.** Chemical structures of representative examples of small molecule USP30 inhibitors are given together with characterization data and references. Cyanamides of covalent inhibitors form isothiourea linkages with the active site cysteine of USP30.

Non-covalent inhibitors belong either to chemical series which were developed from phenylalanine derivatives (with either a benzenesulfonamide or a naphthylsulfonamide) or are natural product derivatives.

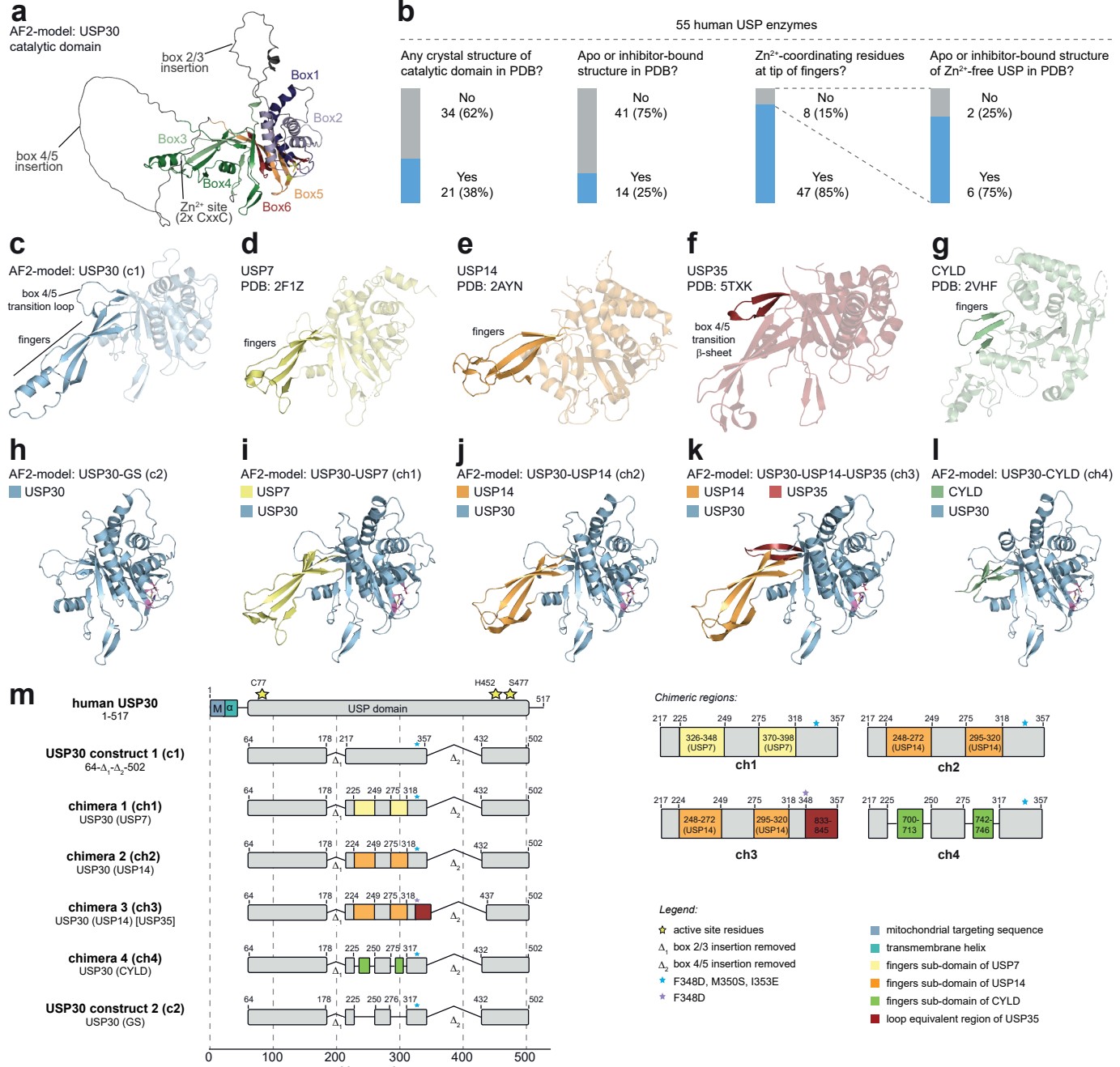

**Extended Data Fig. 2 | Design of chimeric USP30 protein constructs.**
**a**, AlphaFold2 (AF2)-model of the catalytic domain of human USP30. USP boxes and domain insertions are indicated. **b**, Statistics of human USP enzymes regarding their characterization by crystallography and regarding the presence of Zn²⁺-coordinating residues in the tip of the fingers subdomain. **c**, AF2-model of USP30$^{c1}$, previously optimized for structural studies and used as a starting point for this project. **d-g**, Experimental structures of catalytic domains of USP7 (**d**), USP14 (**e**), USP35 (**f**), and CYLD (**g**). Elements used for USP30 chimeric engineering are highlighted. **h-l**, AF2-models of USP30 constructs explored in this study (**h**: USP30-GS (c2), **i**: USP30-USP7 (ch1), **j**: USP30-USP14 (ch2), **k**: USP30-USP14-USP35 (ch3), **l**: USP30-CYLD (ch4)). Catalytic residues are shown in pink. **m**, Architecture of USP30 constructs with close-up view of the boundaries of the chimeric portions.

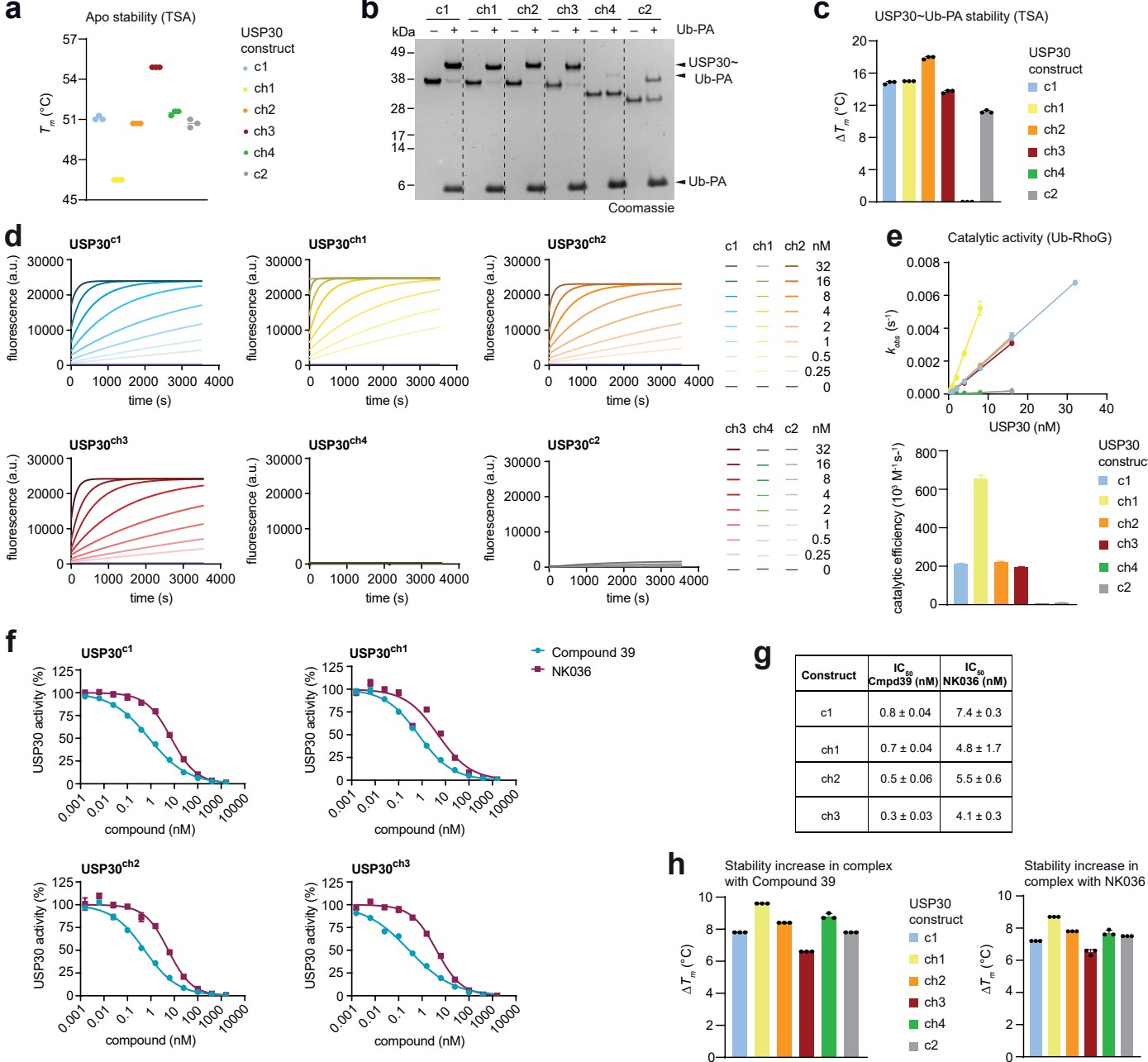

**Extended Data Fig. 3 | Biochemical characterization of USP30 chimeras.**
**a**, Protein stability assessment of USP30 constructs with thermal shift assays. Mean (N = 3 independent replicates). **b**, Ubiquitin probe reactivity assay. Samples were analyzed by SDS-PAGE and Coomassie staining. **c**, Changes in protein stability upon binding to Ub-PA. $\Delta T_m$ was calculated as $T_m$ (Ub-PA-bound) subtracted from $T_m$ (apo protein). Mean ± s.d. (N = 3 independent replicates). **d**, Quantification of enzyme activity. Varying concentrations of USP30 proteins were incubated with Ubiquitin-RhoG substrate and fluorescence was recorded. **e**, Observed rate constants derived from plots in d were plotted over enzyme concentrations (upper panel) to derive catalytic efficiencies (lower panel).

Mean ± s.e.m. (derived from curve fitting, with data for each concentration recorded as N = 3 independent replicates). **f**, Inhibitory potencies of Compound 39 and NK036. Compounds were pre-incubated with USP30 constructs for 1.5 h, and remaining activities were determined from Ub-RhoG cleavage assays. Mean ± s.d. (N = 3 independent replicates). **g**, $IC_{50}$ values of assays shown in f. Mean ± s.e.m. (derived from curve fitting). **h**, Assessment of binding of Compound 39 and NK036 to indicated USP30 constructs by thermal shift assays. $\Delta T_m$ was calculated as $T_m$ of the inhibitor-bound sample subtracted from the $T_m$ of the apo protein. Mean ± s.d. (N = 3 independent replicates).

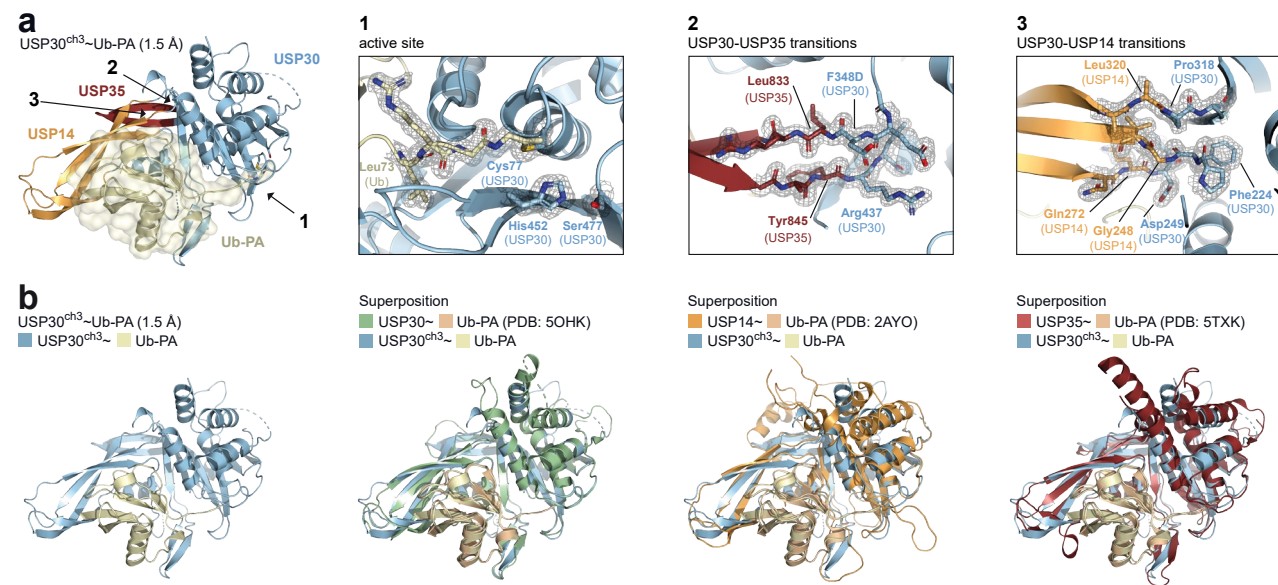

**Extended Data Fig. 4 | Crystal structure of USP30^ch3 in complex with Ub-PA. a**, Cartoon representation of crystal structure of USP30^ch3 - Ub-PA, with representative 2m$F_o$-D$F_c$ electron density (contoured at 1σ) shown for indicated areas. **b**, USP30^ch3 - Ub-PA structure and superpositions with previously reported Ub-PA complexes of USP30^c1, USP14 and USP35.

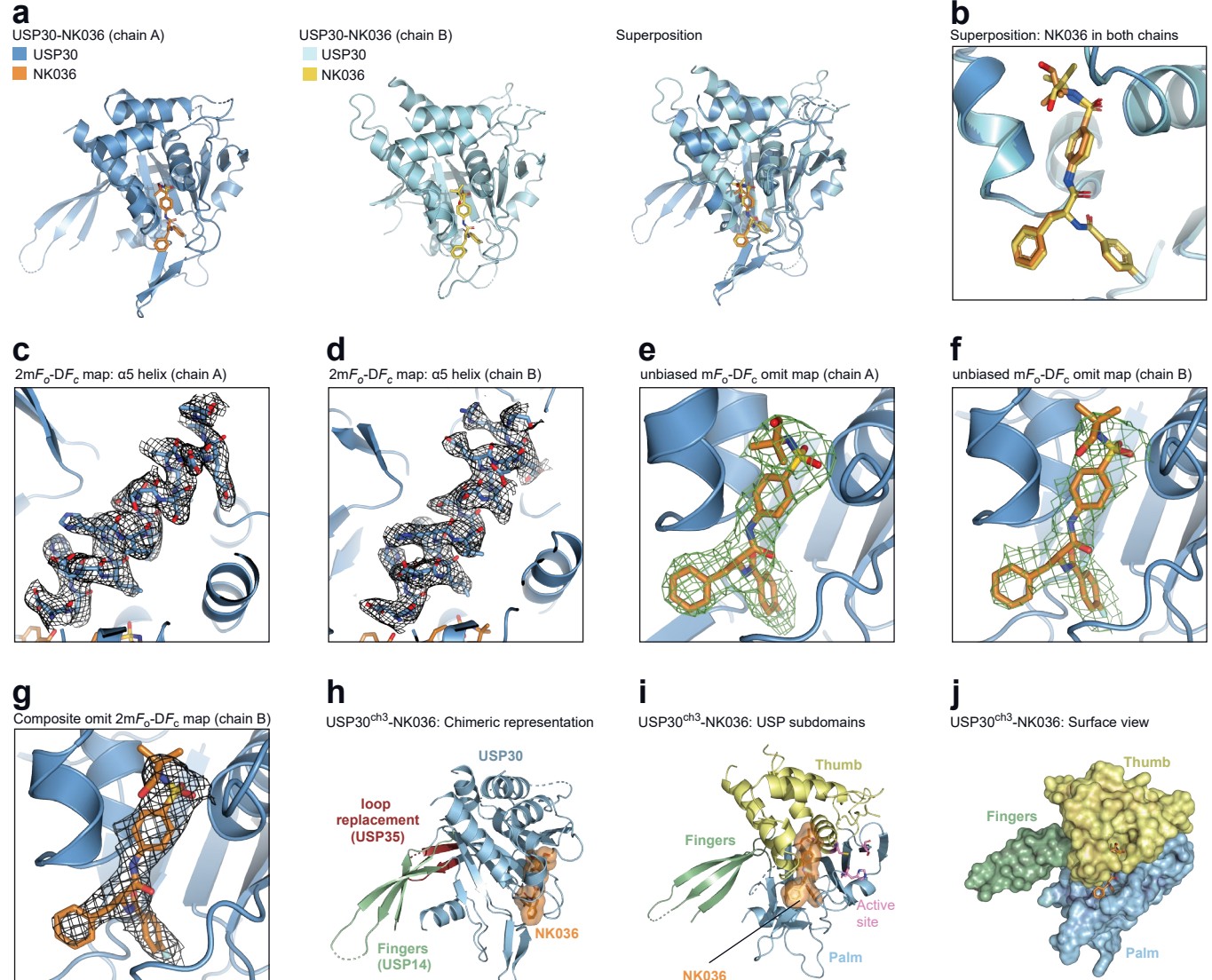

**Extended Data Fig. 5 | Crystal structure of USP30 in complex with NK036. a**, Cartoon representations of the two copies within the asymmetric unit in the crystal structure of USP30^ch3-NK036. **b**, Close-up view of the superposition shown in a highlighting the ligand geometries in both chains. **c-d**, Representative electron density map of the α5 helix in chain A (panel **c**) and chain B (panel **d**). $2mF_o$-D$F_c$ density, contoured at 1σ, with −30 Å$^2$ $B$-factor sharpening, is shown covering all atoms of the region. **e-f**, Unbiased electron density omit map of NK036 in chain A (panel **e**) and chain B (panel **f**). A $mF_o$-D$F_c$ map, contoured at 1σ, covering all atoms of the compound is shown, which was calculated from the

protein geometry before the ligand was modelled. **g**, Composite omit electron density map of NK036 in chain B ($2mF_o$-D$F_c$, contoured at 1σ, covering all atoms of the compound, created as described for Fig. 2b). **h**, Structure as in a, with chimeric elements shown in different colors. Of note, no chimeric residue is near the compound binding site. **i**, Cartoon representation of crystal structure of USP30^ch3-NK036, highlighting different USP subdomains. The compound is shown under an orange surface, active site residues are shown in pink. **j**, Structure as in panel e with surface representation of USP30 highlighting different USP subdomains.

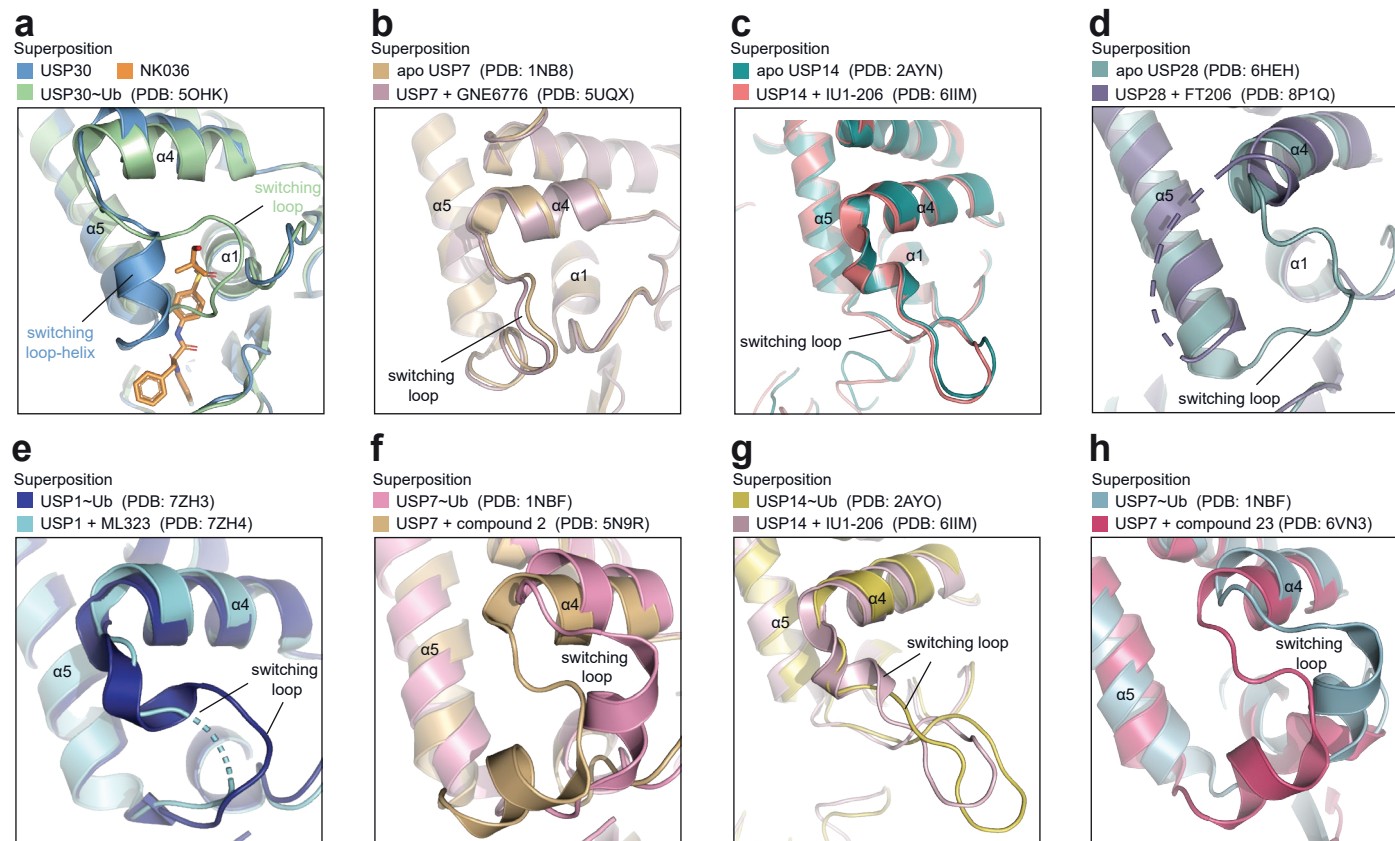

**Extended Data Fig. 6 | Conformational change of switching loop upon compound binding. a-h,** Structural superpositions showing the switching loop positions in different USP DUBs, comparing compound-bound to ubiquitin-bound or apo states (**a**: USP30 with NK036 compared to Ubiquitin-bound USP30, **b**: USP7 with GNE6776 compared to apo USP7, **c**: USP14 with IU1-206 compared to apo USP14, **d**: USP28 with FT206 compared to apo USP28, **e**: USP1 with ML323 compared to Ubiquitin-bound USP1, **f**: USP7 with compound 2 compared to Ubiquitin-bound USP7, **g**: USP14 with IU1-206 compared to Ubiquitin-bound USP14, **h**: USP7 with compound 23 compared to Ubiquitin-bound USP7).

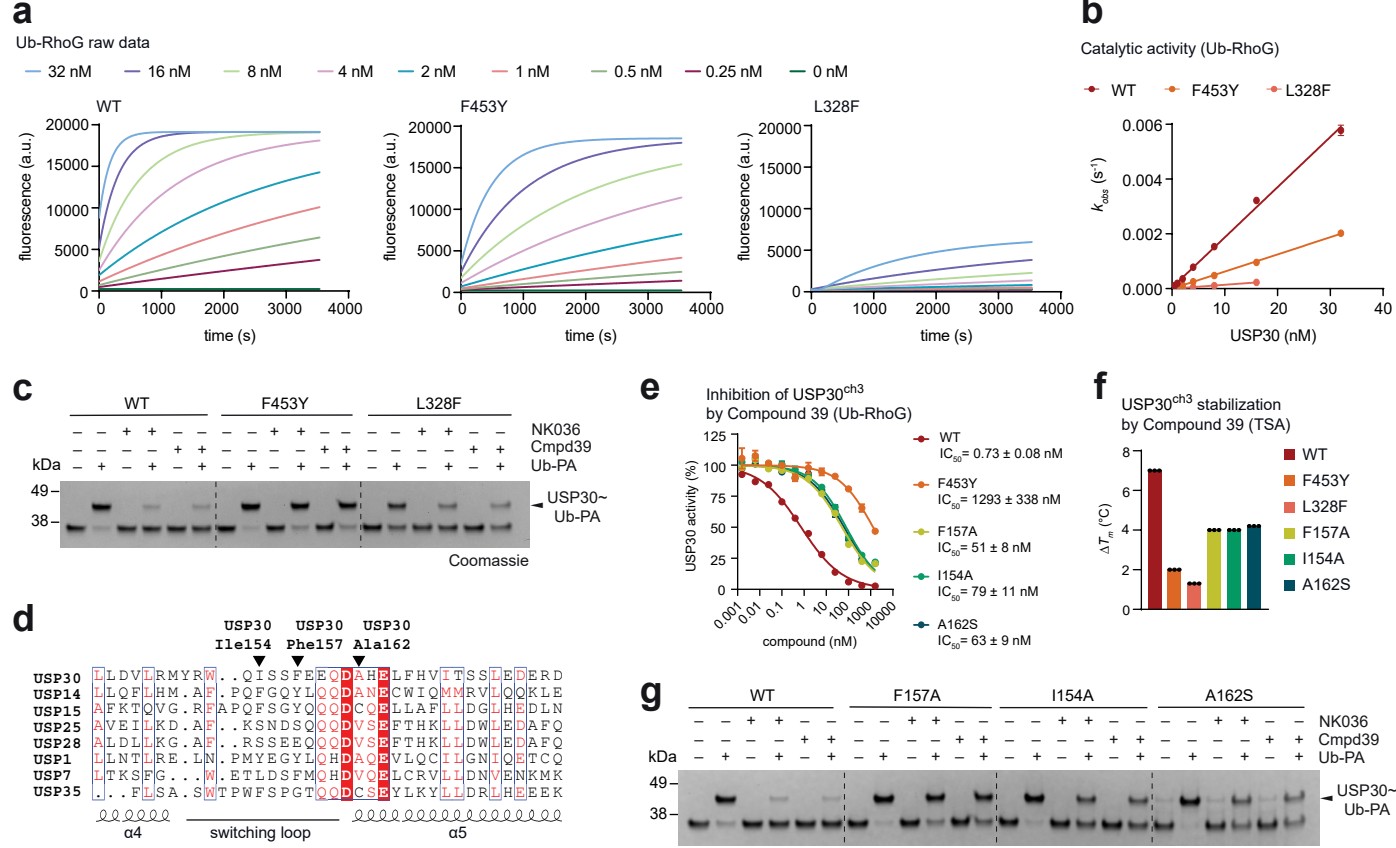

**Extended Data Fig. 7 | Molecular basis of compound potency and specificity towards USP30. a-b**, Quantification of USP30 enzyme activity for WT and indicated mutant proteins (**a**: fluorescence over time plots for each protein, **b**: observed rate constant over enzyme concentration plots), as described in Extended Data Fig. 3d, e. **c**, Probe competition assay. USP30 and inhibitors were preincubated, followed by addition of Ub-PA, and analysis of samples by SDS-PAGE and Coomassie staining. **d**, Sequence alignment of indicated human USP DUBs. Arrows indicate Ile154, Phe157 and Ala162 residues in USP30.

**e**, Inhibitory potencies of Compound 39, pre-incubated with indicated USP30 proteins for 1.5 h, determined from Ub-RhoG cleavage assays. IC$_{50}$ values are given as mean ± s.e.m. (derived from curve fitting, with data for each concentration recorded as N = 3 independent replicates). **f**, Protein stability of indicated USP30 proteins in the presence of 20 µM Compound 39. $\Delta T_m$ was calculated as $T_m$ of the compound-bound sample subtracted from $T_m$ of the respective apo protein. Mean ± s.d. (N = 3 independent replicates). **g**, Probe competition assay performed as in c with mutations characterized in e and f.

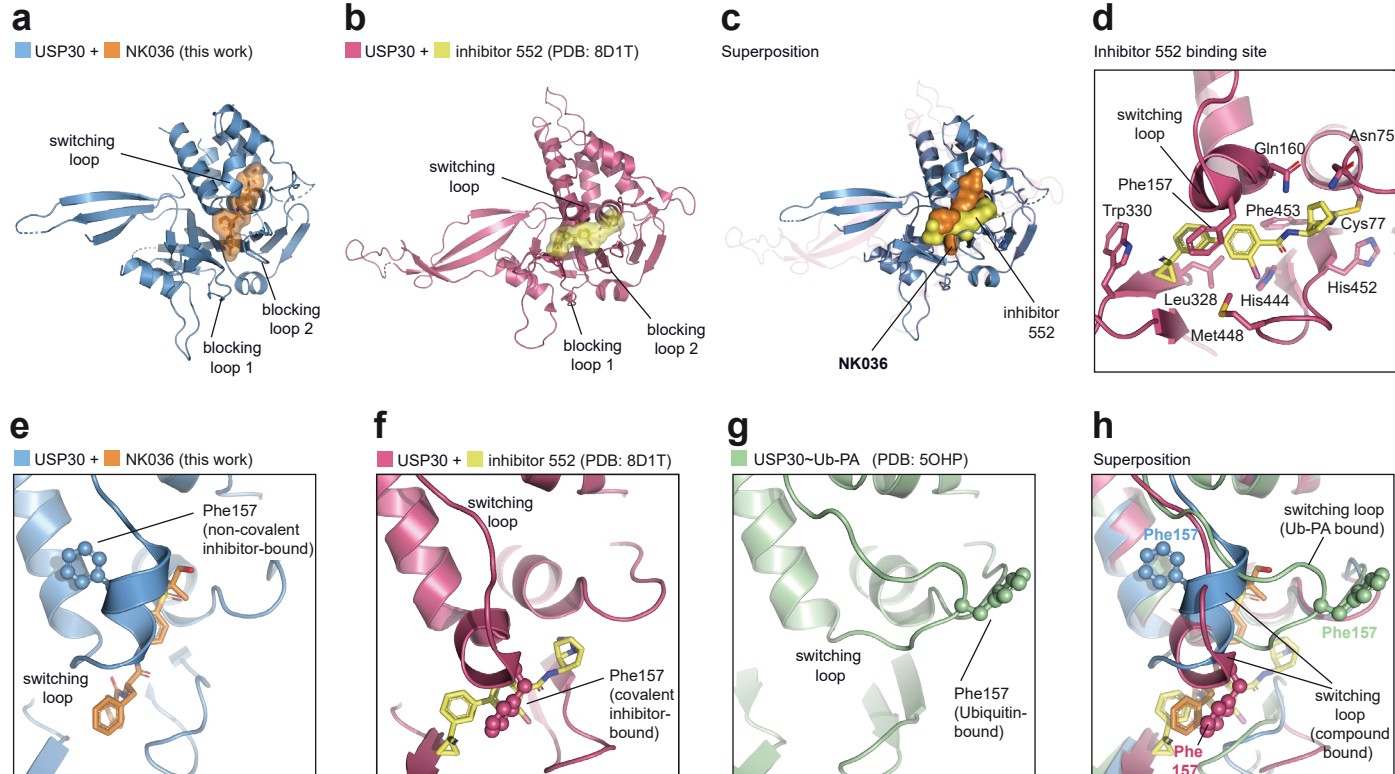

**Extended Data Fig. 8 | Comparison of the binding modes of covalent and non-covalent USP30 inhibitors. a-b**, Cartoon representations of the crystal structures of USP30 in complex with non-covalent inhibitor NK036 (**a**) and covalent inhibitor 552 (**b**). The fragment antigen-binding region (Fab) present in the structure 8D1T is not shown. **c**, Superposition of the structure of USP30 + NK036 on the structure of USP30-inhibitor 552. **d**, Close-up view of the inhibitor 552 binding site highlighting key residues. The chemical structure of inhibitor 552 is shown in Extended Data Fig. 1. **e-h**, Close-up views of the conformational changes of the switching loop in the structures of USP30 + NK036 (**e**), USP30-inhibitor 552 (**f**) and USP30-Ub-PA (**g**). A superposition of these three structures is also shown (**h**). Highlighted are different orientations of the switching loop residue Phe157 in the respective structures. Upon binding of the non-covalent compound, Phe157 moves backwards leading to the formation of the cryptic pocket. In contrast, upon binding of the covalent compounds, Phe157 moves forward which creates a shielded tunnel between the modified catalytic cysteine and the S1 Ubiquitin site.

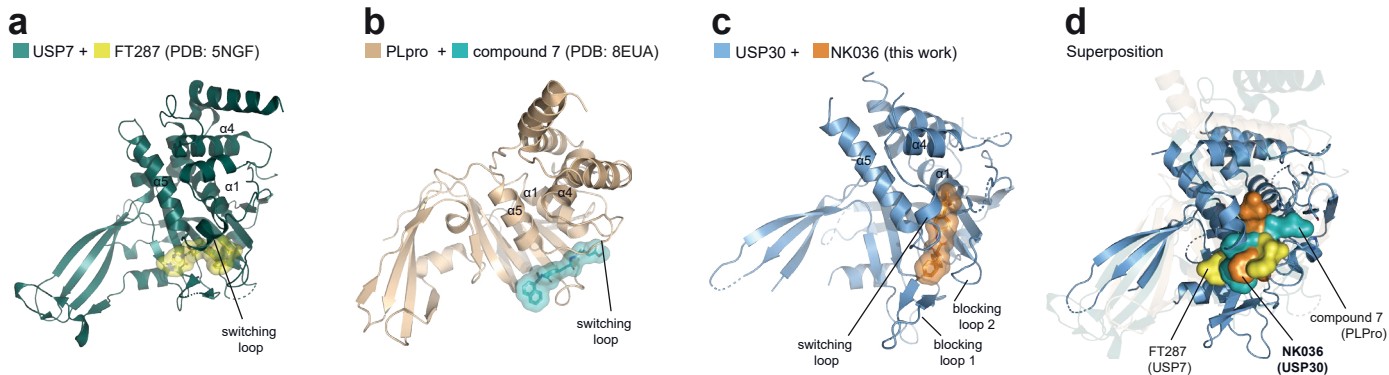

**Extended Data Fig. 9 | Comparison of the binding mode of NK036 to covalent USP DUB inhibitors. a-c**, Cartoon representations of the crystal structures of USP7 (**a**), SARS-CoV2-PLPro (**b**) and USP30 (**c**) in complex with respective inhibitors. Compounds are shown as surfaces. **d**, Superposition of the structure of USP30 + NK036 on other structures shown in panels **a-c**. Compounds are shown as surfaces and are labeled. All protein cartoons except USP30 are semitransparent.

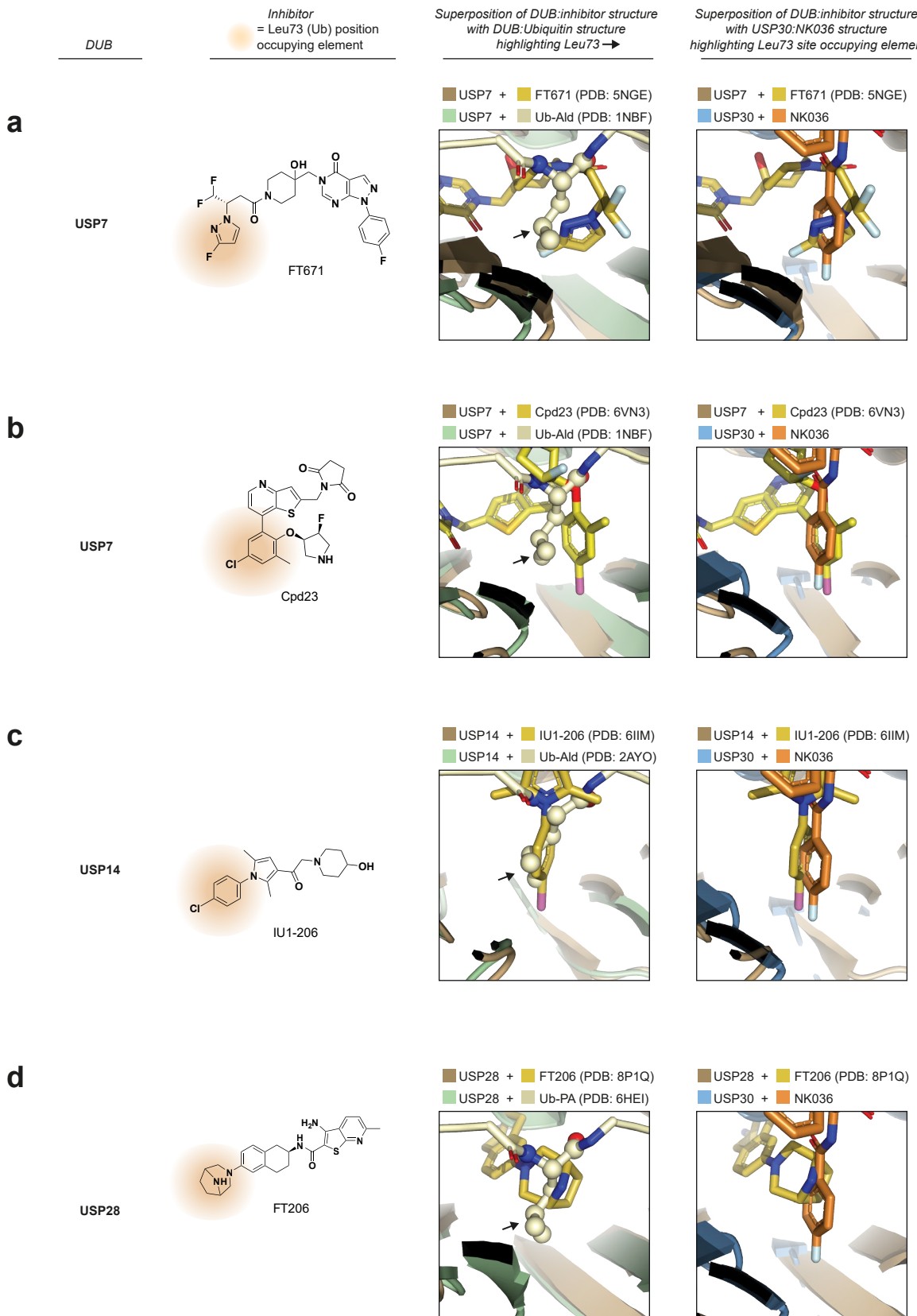

**Extended Data Fig. 10 | A common DUB ligandability hotspot and a shared hotspot anchor. a-d,** Chemical and structural representations of inhibitors of USP7 (**a, b**), USP14 (**c**) and USP28 (**d**). Shown are the overlays of the inhibitor-bound structures with ubiquitin-bound structures of the respective DUBs (first box) and the inhibitor-bound structures of each DUB overlaid on NK036 bound structure of USP30 (second box). Highlighted is the close-up view of the compound binding site at the ubiquitin Leu73 pocket. Please note the chemically related hotspot anchors in **b** and **c**. Structural superpositions were aligned taking only the protein residues into account. The chemical moieties of the inhibitors that occupy the Leu73 pocket are highlighted with orange background.

# Reporting Summary

## Statistics

For all statistical analyses, confirm that the following items are present in the figure legend, table legend, main text, or Methods section.

| n/a | Confirmed | |
|---|---|---|
| ☐ | ☒ | The exact sample size (*n*) for each experimental group/condition, given as a discrete number and unit of measurement |
| ☒ | ☐ | A statement on whether measurements were taken from distinct samples or whether the same sample was measured repeatedly |
| ☒ | ☐ | The statistical test(s) used AND whether they are one- or two-sided *Only common tests should be described solely by name; describe more complex techniques in the Methods section.* |
| ☒ | ☐ | A description of all covariates tested |
| ☒ | ☐ | A description of any assumptions or corrections, such as tests of normality and adjustment for multiple comparisons |
| ☐ | ☒ | A full description of the statistical parameters including central tendency (e.g. means) or other basic estimates (e.g. regression coefficient) AND variation (e.g. standard deviation) or associated estimates of uncertainty (e.g. confidence intervals) |
| ☒ | ☐ | For null hypothesis testing, the test statistic (e.g. *F*, *t*, *r*) with confidence intervals, effect sizes, degrees of freedom and *P* value noted *Give P values as exact values whenever suitable.* |
| ☒ | ☐ | For Bayesian analysis, information on the choice of priors and Markov chain Monte Carlo settings |
| ☒ | ☐ | For hierarchical and complex designs, identification of the appropriate level for tests and full reporting of outcomes |
| ☒ | ☐ | Estimates of effect sizes (e.g. Cohen's *d*, Pearson's *r*), indicating how they were calculated |

*Our web collection on statistics for biologists contains articles on many of the points above.*

## Software and code

Policy information about availability of computer code

| Data collection | SparkControl (Tecan, V2.3); CFX Maestro (Bio-Rad, V4.1.2433.1219); OpenLab CDS (Agilent, V2.4) |
|---|---|
| Data analysis | Image Lab (Bio-Rad, V2.4.0.03); Prism (GraphPad, V9); Pymol (Schrödinger, V2.5.5); Dials (V3.19.0 and V3.16.0); Aimless (V0.7.15); Staraniso (V3.350); MR Phaser (V2.8.3); Coot (V0.9.8.93); Phenix.Refine (V1.20.1); ORCA (V5.0.4); Alphafold (V2.3.0); eLBOW (V1.20.1) |

For manuscripts utilizing custom algorithms or software that are central to the research but not yet described in published literature, software must be made available to editors and reviewers. We strongly encourage code deposition in a community repository (e.g. GitHub). See the Nature Portfolio guidelines for submitting code & software for further information.

## Data

Policy information about availability of data

All manuscripts must include a data availability statement. This statement should provide the following information, where applicable:
- Accession codes, unique identifiers, or web links for publicly available datasets
- A description of any restrictions on data availability
- For clinical datasets or third party data, please ensure that the statement adheres to our policy

Coordinates and structure factors for the USP30ch3 + NK036 and USP30ch3~Ub-PA crystal structures were deposited with the protein databank (PDB) under accession codes 9F19 and 9F6G, respectively. Coordinates of other structures were obtained from the PDB through accession codes 2AYN, 5N9R, 5OHK, 5OHP, 5TXK, 5UQX, 6GH9, 6IIM, 6IIN, 6VN3, 7ZH4, 8P1Q. Source data (uncropped gels and blots, numerical data) are provided with this paper.

# Research involving human participants, their data, or biological material

Policy information about studies with <u>human participants or human data</u>. See also policy information about <u>sex, gender (identity/presentation), and sexual orientation</u> and <u>race, ethnicity and racism</u>.

| | |
|---|---|
| Reporting on sex and gender | Human research participants were not involved in this study. |
| Reporting on race, ethnicity, or other socially relevant groupings | Human research participants were not involved in this study. |
| Population characteristics | Human research participants were not involved in this study. |
| Recruitment | Human research participants were not involved in this study. |
| Ethics oversight | Human research participants were not involved in this study. |

Note that full information on the approval of the study protocol must also be provided in the manuscript.

# Field-specific reporting

Please select the one below that is the best fit for your research. If you are not sure, read the appropriate sections before making your selection.

☒ Life sciences  ☐ Behavioural & social sciences  ☐ Ecological, evolutionary & environmental sciences

For a reference copy of the document with all sections, see <u>nature.com/documents/nr-reporting-summary-flat.pdf</u>

# Life sciences study design

All studies must disclose on these points even when the disclosure is negative.

| | |
|---|---|
| Sample size | For quantitative experiments, a sample size of 3 to 6 independent experiments was chosen in line with what is the standard of the field in the molecular biosciences. Sample sizes are given in the figure captions. |
| Data exclusions | Data exclusion occurred in the processing of crystallographic data as implemented in the respective software (e.g. in Phenix.Refine during the scaling of input intensities and subsequent outlier rejection according to expected intensity statistics). These processes took place completely automated as is the default in these programs and without any customization or user input. |
| Replication | All observations were made in at least two independent experiments, typically with technical triplicates, all with consistent results. The numbers of independent experiments is given and individual results are shown were possible. |
| Randomization | Randomization was not applicable as no experiments involving humans/animals were performed. Importantly, no experiment was sensitive to the order of measurement/treatment. |
| Blinding | Blinding was not carried out as no subjective analysis (e.g. scoring) was performed. |

# Reporting for specific materials, systems and methods

We require information from authors about some types of materials, experimental systems and methods used in many studies. Here, indicate whether each material, system or method listed is relevant to your study. If you are not sure if a list item applies to your research, read the appropriate section before selecting a response.

## Materials & experimental systems

| n/a | Involved in the study |
|---|---|
| ☐ | ☒ Antibodies |
| ☐ | ☒ Eukaryotic cell lines |
| ☒ | ☐ Palaeontology and archaeology |
| ☒ | ☐ Animals and other organisms |
| ☒ | ☐ Clinical data |
| ☒ | ☐ Dual use research of concern |
| ☒ | ☐ Plants |

## Methods

| n/a | Involved in the study |
|---|---|
| ☒ | ☐ ChIP-seq |
| ☒ | ☐ Flow cytometry |
| ☒ | ☐ MRI-based neuroimaging |

# Antibodies

| Antibodies used | Primary antibodies:<br>anti-Flag, 1:1000, Sigma, F3165 (clone M2)<br>anti-USP30, 1:500, Sigma, HPA016952 (polyclonal)<br>anti-Vinculin, 1:10000, Sigma, V9131 (clone hVIN-1)<br>anti-GAPDH, 1:10000, Thermo Fisher, AM4300 (clone 6C5)<br>anti-TOM40, 1:2000, Proteintech, 18409-1-AP (polyclonal)<br>anti-TOM20, 1:5000, Proteintech, 11802-1-AP (polyclonal)<br>anti-MFN2, 1:5000, Proteintech, 12186-1-AP (polyclonal)<br><br>Horseradish peroxidase-coupled secondary antibodies:<br>anti-mouse, 1:5000, Sigma, NXA931<br>anti-rabbit, 1:5000, Sigma, GENA934 |
| --- | --- |
| Validation | All primary antibodies are validated for western blotting on human cell lysates:<br><br>anti-Flag: "Monoclonal ANTI-FLAG® M2 antibody produced in mouse has been used in: immunoblotting. // technique(s): western blot: 10 µg/mL" https://www.sigmaaldrich.com/DE/en/product/sigma/f3165<br><br>anti-USP30: "All Prestige Antibodies Powered by Atlas Antibodies are developed and validated by the Human Protein Atlas (HPA) project and as a result, are supported by the most extensive characterization in the industry. // technique(s): immunoblotting: 0.04-0.4 µg/mL" https://www.sigmaaldrich.com/DE/en/product/sigma/hpa016952<br><br>anti-Vinculin: "Monoclonal Anti-Vinculin antibody produced in mouse has been used in: western blotting. // technique(s): western blot: 1:200 using extract of human fibroblasts" https://www.sigmaaldrich.com/DE/en/product/sigma/v9131<br><br>anti-GAPDH: "Applications: Western Blot (WB), Species Reactivity: Amphibian, Dog, Chicken, Fish, Human, Mouse, Non-human primate, Rabbit, Rat" https://www.thermofisher.com/antibody/product/GAPDH-Antibody-clone-6C5-Monoclonal/AM4300<br><br>anti-TOM40: "KD/KO Validated" "Positive WB detected in HEK-293 cells, HeLa cells" https://www.ptglab.com/products/TOMM40-Antibody-18409-1-AP.htm<br><br>anti-TOM20: "KD/KO Validated" "Positive WB detected in HEK-293 cells, HeLa cells" https://www.ptglab.com/products/TOM20-Antibody-11802-1-AP.htm<br><br>anti-MFN2: "KD/KO Validated" "Positive WB detected in mouse brain tissue, HeLa cells" https://www.ptglab.com/products/MFN2-Antibody-12186-1-AP.htm |

# Eukaryotic cell lines

Policy information about cell lines and Sex and Gender in Research

| Cell line source(s) | HEK293 cells were purchased from the DSMZ repository (DSMZ no: ACC 305). HeLa cells stably expressing YFP-Parkin were kindly provided by Dr. Richard Youle (National Institute of Neurological Disorders and Stroke, Bethesda, Maryland, U.S.A.). |
| --- | --- |
| Authentication | Cells were used without authentication. |
| Mycoplasma contamination | Cells were tested for mycoplasma contamination with a negative result. |
| Commonly misidentified lines (See ICLAC register) | No commonly misidentified lines were used in this study. |

# Plants

| Seed stocks | No plants were involved in this study |
| --- | --- |
| Novel plant genotypes | No plants were involved in this study |
| Authentication | No plants were involved in this study |

