## [Peer Review File · Nature Structural & Molecular Biology]

Chimeric deubiquitinase engineering reveals structural basis for specific inhibition of the mitophagy regulator USP30

Corresponding Author: Dr Malte Gersch

A version of this paper was originally rejected for publication by Nature Structural & Molecular Biology, however that decision was reconsidered after appeal by the authors.

Version 0:

Decision Letter:

11th Jul 2024

Dear Dr. Gersch,

Thank you for submitting your manuscript "Chimeric deubiquitinase engineering reveals structural basis for specific inhibition of USP30". We have now carefully evaluated the work and discussed it among the editorial team. Unfortunately, we have decided not to consider the manuscript further for publication in Nature Structural & Molecular Biology, unless the physiological relevance of the findings is validated with cellular data or the generalisability of the Leu73-Ubiquitin binding site is extensively consolidated.

We can only consider a small proportion of the manuscripts submitted to our journal and are often forced to make difficult decisions. Manuscripts are evaluated editorially for their potential interest to a broad audience, the level of novel insight obtained and whether the findings represent a significant advance relative to the published literature, among other considerations.

In this case, we are interested in studies that progress our molecular understanding of USP30 inhibition, particularly when these principles are validated in cells or in vivo. We appreciate the intelligent strategy employed to obtain the USP30-inhibitor structures and the development of NK036, as well the identification of a cryptic pocket and the potential implications of a common ligandability hotspot centered on Leu73. We recognise that these findings will be of value to others working in this area. However, after discussion among the editorial staff, I am afraid we are not persuaded that the level of biological and functional insight obtained warrants sending the manuscript out for review in Nature Structural & Molecular Biology, in the absence of cellular data, akin to those in publications 23,25,27 (for example cellular assays like those in Fig.3,4 in reference 27, or figures 4-6 in reference 25, or Fig.3,4 in reference 23), or others, validating the physiological relevance of the findings. We deem that this is particular important for a newly developed compound like NK036. Generating a reasonable amount of such cellular data (at least a main figure, in which also the F453Y and L328F mutants are also interrogated and the effects on mitophagy are ascertained) or providing strong evidence for the generalisability of the ligandability principles on Leu73 of ubi would make this a much stronger candidate for peer review at NSMB.

If obtaining such data and resubmitting to NSMB is something that would interest you, please do not hesitate to write me. If however, these seem out of scope in your opinion, my colleagues at Nature Communications will send your manuscript out for external review. To transfer your manuscript please use our manuscript transfer portal. You will not have to re-supply manuscript metadata and files, unless you wish to make modifications. For more information, please see our [manuscript transfer FAQ](http://www.nature.com/authors/author_resources/transfer_manuscripts.html?WT.mc_id=EMI_NPG_1511_AUTHORTRANSF&WT.ec_id=AUTHOR) page.

I am sorry we could not be more positive on this occasion. We thank you for the opportunity to consider this work and wish you success in seeking publication elsewhere.

Sincerely,

Dimitris Typas
Senior Editor
Nature Structural & Molecular Biology
ORCID: 0000-0002-8737-1319

** For Springer Nature Limited general information and news for authors, see <http://npg.nature.com/authors>.

Version 1:

Decision Letter:

10th Oct 2024

Dear Dr. Gersch,

Thank you again for submitting your manuscript "Chimeric deubiquitinase engineering reveals structural basis for specific inhibition of USP30". I apologise for the delay in returning our editorial decision, which resulted from the difficulty in timely obtaining suitable referee reports and adequately discussing them within the editorial team. Nevertheless, we now have comments (below) from the 3 reviewers who evaluated your paper. In light of these reports, we remain interested in your study and would like to see your response to the comments of the referees, in the form of a revised manuscript.

You will see that the reviewers appreciate the potential of the study, both in terms of the progress it imparts on USP30 and in terms of the developed approach and its application to other DUBs. Nevertheless, they raise concerns that must be addressed in a revised manuscript. More specifically, the referees request clarifications, missing controls, additional analyses, missing information (such as Fo–Fc omit maps), and additional biochemical experiments that may boost the potential for inhibitor design. Furthermore, our editorial view remains that inclusion of cellular data, demonstrating the implications of the mechanistic insight in mitophagy, would significantly increase the impact and reach of the paper to the broad audience of NSMB readers. Please be sure to address/respond to all concerns of the referees in full in a point-by-point response and highlight all changes in the revised manuscript text file. If you have comments that are intended for editors only, please include those in a separate cover letter.

We expect to see your revised manuscript within 3-6 months. If you cannot send it within this time, please contact us to discuss an extension; we would still consider your revision, provided that no similar work has been accepted for publication at NSMB or published elsewhere.

Reporting Summary:

If there are additional or modified structures presented in the final revision, please submit the corresponding PDB validation reports, as well as relevant maps and models.

SOURCE DATA: we urge authors to provide, in tabular form, the data underlying the graphical representations used in figures. This is to further increase transparency in data reporting, as detailed in this editorial (<http://www.nature.com/nsmb/journal/v22/n10/full/nsmb.3110.html>). Spreadsheets can be submitted in excel format. Only

one (1) file per figure is permitted; thus, for multi-paneled figures, the source data for each panel should be clearly labeled in the Excel file; alternately the data can be provided as multiple, clearly labeled sheets in an Excel file. When submitting files, the title field should indicate which figure the source data pertains to. We encourage our authors to provide source data at the revision stage, so that they are part of the peer-review process.

Data availability: this journal strongly supports public availability of data. All data used in accepted papers should be available via a public data repository, or alternatively, as Supplementary Information. If data can only be shared on request, please explain why in your Data Availability Statement, and also in the correspondence with your editor. Please note that for some data types, deposition in a public repository is mandatory - more information on our data deposition policies and available repositories can be found below:

<https://www.nature.com/nature-research/editorial-policies/reporting-standards#availability-of-data>

Link Redacted

Sincerely,

Dimitris Typas
Senior Editor
Nature Structural & Molecular Biology
ORCID: 0000-0002-8737-1319

Reviewers' Comments:

Reviewer #1 (Remarks to the Author):

Kazi et al., report the crystal structure of USP30 in complex with a small molecule inhibitor. Despite USP30 being a prime drug target for therapeutics and USP30 inhibitors undergoing clinical trials for Parkinson's disease, the structural bases for USP30 small molecule inhibition are unclear. Some prior information was eluded by O'brien et al., 2023 (<https://pubmed.ncbi.nlm.nih.gov/37385347/>), and this has been incorporated in the current study design.

The novelty of the manuscript stems from engineering a USP30 construct that is readily crystallisable with a small-molecule ligand, which has so far been difficult. This comes from the author's prior knowledge and expertise on USP30 structure and function including their HDX analysis showing mobile USP30 regions. In the current study, the authors have switched these flexible regions with more 'stable' regions that were grafted onto the USP30 catalytic domain, thus generating a USP30 chimera that can be used for crystallisation. The manuscript has ample data to support that the chimeric protein is stable, sensitive to small molecule inhibitors and binds ubiquitin, and the authors have gone to great lengths to justify their use of this approach. As such the manuscript represents the generation of a highly valuable tool for advancing USP30 small-molecule inhibitor efforts. Analysis of their crystal structure bound to a USP30 inhibitor has also generated insights into druggable pockets of USP DUBs. This is useful in explaining the current small molecule inhibitor mechanism of action and can aid future inhibitor design.

There is a PDB entry (8D1T and related 8D0A) showing the structure of USP30 bound to a covalent small molecule inhibitor and nanobody, although no paper has been published, which highlights the importance of this manuscript. Moreover, the strategy described here will dispense from the use of a nanobody.

While some of the data and approach may appear lacking in novelty (the authors didn't discover new inhibitors or an unexpected inhibitor binding site), on balance, this reviewer thinks that the technical advancement in engineering a new construct and the comprehensive structure analyses merit publication in NSMB. This strategy will be useful to many in the field and can become standard practice for those interested in developing small-molecule inhibitors of USP30 and other DUBs.

Minor comments:

-- R-free is on the high side for the USP30-inhibitor structure (26.6%). Can the authors improve this or comment on why this is high?

-- Omit maps after simulated annealing or unbiased Fo-Fc maps (before the ligand was modelled and refined) should be shown in Figures 2B and Ext. Data Fig. 5C

-- Can the authors compare USP30 structures with covalent inhibitors (PDBid 8D1T; 8D0A) to their own? Are there major differences / commonalities worth commenting on?

-- How do USP30 (ch3) crystal structures compare to an AlphaFold 3 predicted model? What is the RMSD? Will be useful to compare these and perhaps comment on the ability of AF3 to predict chimeric proteins.

Reviewer #2 (Remarks to the Author):

USP30 is a member of a Ub-specific protease (USP) family and is localized on mitochondria. USP30 regulates the basal level of mitochondrial ubiquitination and opposes Parkin/Pink1-mediated mitophagy, depending on its deubiquitinating activity. The inhibition of USP30 promotes the clearance of damaged mitochondria. Therefore, inhibitors of USP30 have the potential to treat diseases caused by dysfunctional mitochondria, like Parkinson's disease. To date, several scaffolds of small molecule USP30 inhibitors have been reported, but the structural basis for their inhibition mechanisms remains unknown. In this paper, Haque Kazi et al. determined the crystal structure of the engineered human USP30 in complex with a specific non-covalent inhibitor. The structure and structure-based site-directed mutagenesis study elucidated how this compound specifically inhibits USP30 at the atomic level. The chimeric engineering strategy for improving the crystallizability of USP family enzymes may be useful for structure-based drug development targeting this enzyme family. Overall, every experiment was appropriately performed, and the result is explicitly presented, although there is a concern regarding the diffraction anisotropy of the inhibitor-bound structure. The findings including an identification of the unexplored cryptic ligand binding pocket are novel and may attract considerable interest to readers, particularly in the research field for the biological system associated with ubiquitin and autophagy. I suggest the following points to be addressed before publication.

Major points:

1. Please provide the statistical information of the inner shell (i.e., at the resolution limit along the lowest resolution axis) in the diffraction anisotropy analysis by STARANISO. Spherical data completeness in the inner shell is important to assess the data quality. Also, please transform the crystal lattice axis so that the space group becomes P21212 if possible. P21221 may be unconventional for protein crystallography.
2. An Fo-Fc omit map should also be shown for the bound inhibitors. I prefer an Fo-Fc omit map to a 2Fo-Fc composite omit map to show the quality of the ligand density. In addition, 2Fo-Fc density maps in a few regions other than the ligand should also be presented to guarantee the data quality. I am not sure whether the diffraction anisotropy correction worked fine or not from the data presented in the current manuscript.

Minor points:

1. What does "MOMP" indicate in Fig. 1a? Mitochondrial outer membrane proteins?
2. "AlphaFold-based modeling" may be "AlphaFold2-based modeling". As far as I know, the architecture of AlphaFold2 is substantially different from that of the initial version of AlphaFold.
3. "F" that indicates structure factors should be italicized and the subsequent "o" (meaning "observed") and "c" (meaning "calculated") should be subscripts.
4. In Fig. 2b, the color of blocking loop 2 is similar to that of other regions and difficult to distinguish.

Reviewer #3 (Remarks to the Author):

In this study the authors design and evaluate chimeric USP30 constructs in order to increase their ability to obtain co-crystal structures with USP30 inhibitors. Utilising this strategy they are then able to demonstrate for the first time the binding mode of a close analogue of selective reversible USP30 inhibitor 'compound 39', in this case their analogue is NK-036. The co-

crystal structures obtained reveal that the para-Fluoro-benzoyl functionality binds deep in a conserved USP pocket (next to Leu73) which when compared to other known DUB inhibitors suggests this to be a highly ligandable binding site from which future Dub inhibitors could be anchored. Intriguingly, they also show that the rest of the molecule binds in a cryptic pocket that is induced via ligand binding. This is an important, well presented and well supported study that ignites future opportunities and new understanding for DUB inhibitor design and discovery, particular for selective and reversible DUB inhibitor discovery which has been highly challenging. The data is very sound and appears very robust. I would strongly recommend this for publication with only very minor corrections / non-essential suggestions and congratulate the authors on a fantastic piece of work that the community will I'm enjoy reading and find very useful.

Minor correction and suggestion:

- Bottom of page 2, line 58, the sentence starting "Two structurally yet undisclosed..." does not make sense/convey the meaning I think the authors intend and needs to be rephrased.
- These reversible USP30 inhibitors are not particularly drug-like in nature. The specificity for USP30 is described here as arising from the binding mode within the Leu73 pocket. Although this shown to be a ligandable site for multiple DUBs, the F435Y switch in particular supports that this is a key driver in specificity over other DUBs. It would therefore be interesting to observe what contribution is made to affinity and specificity of the rest of the molecule (i.e. the sulfonamide portion). i.e. if one takes the fragment of the portion of the molecule that binds in the Leu73 pocket only, is their measurable affinity to this construct and what is the delta versus the full molecule? This may act as an anchoring fragment for future, more druglike reversible USP30 inhibitor design, and support those suggestions put forward by the authors in the discussion.

Version 2:

Decision Letter:

Our ref: NSMB-A49483B

16th Dec 2024

Dear Dr. Gersch,

Thank you for submitting your revised manuscript "Chimeric deubiquitinase engineering reveals structural basis for specific inhibition of USP30" (NSMB-A49483B). It has now been seen by the original referees and their comments are below. The reviewers find that the paper has improved in revision, and therefore we are happy to accept it in principle in Nature Structural & Molecular Biology, pending minor revisions to comply with our editorial and formatting guidelines.

We are now performing detailed checks on your paper and will send you a checklist detailing our editorial and formatting requirements in 3 weeks. Please do not upload the final materials and make any revisions until you receive this additional information from us.

To facilitate our work at this stage, it is important that we have a copy of the main text as a word file. If you could please send along a word version of this file as soon as possible, we would greatly appreciate it; please make sure to copy the NSMB account (cc'ed above).

Sincerely,

Dimitris Typas
Senior Editor
Nature Structural & Molecular Biology
ORCID: 0000-0002-8737-1319

Reviewer #1 (Remarks to the Author):

The authors have addressed my comments in full, and I support publication.

Reviewer #2 (Remarks to the Author):

NSMB-A49483B

Review comment:

Haque Kazi et al., "Chimeric deubiquitinase engineering reveals structural basis for specific inhibition of USP30"

All my concerns have been properly addressed in the revised manuscript. The current version appears ready for publication.

Reviewer #3 (Remarks to the Author):

Many thanks to the authors for their thorough responses. I congratulate them on their work and recommend for publication.

Version 3:

Decision Letter:

12th Mar 2025

Dear Dr. Gersch,

We are now happy to accept your revised paper "Chimeric deubiquitinase engineering reveals structural basis for specific inhibition of the mitophagy regulator USP30" for publication as an Article in Nature Structural & Molecular Biology.

Your paper will be published online soon after we receive proof corrections and will appear in print in the next available issue. You can find out your date of online publication by contacting the production team shortly after sending your proof corrections.

Sincerely,

Dimitris Typas
Senior Editor
Nature Structural & Molecular Biology
ORCID: 0000-0002-8737-1319

We would like to sincerely thank all reviewers for the careful analysis of our work and for their constructive suggestions. We were truly thrilled to read their supportive comments. We have addressed all raised points in full through additional analyses and experiments and hope that the manuscript is now judged ready for publication.

We would also like to thank the editor for the support for our work and for the suggestion to strengthen the manuscript through exploring how the here characterized USP30 inhibitor and associated inhibition-resistant mutants can be investigated in a cellular context. To this end, we first introduced three additional compound-resistant mutations. These mutations cover critical residues of the inhibitor binding site: Phe157 (on the switching loop, its large conformation change is described in Fig. 3), Ala162 (the gatekeeper residue of the cryptic pocket) and Ile154 (in the upper part of the thumb domain, contacting the sulfonamide).

Corresponding data are shown in Fig. 5d-f. These mutations have higher IC_{50} values, show less protein stabilization upon inhibitor binding and, contrary to wild-type USP30, are amenable to ubiquitin probe labeling when intact cells were treated with NK036. Additional data are presented in Extended Data Fig. 7 (kinetic assay raw data, sequence alignment, in vitro probe assays).

These data are complementary to the mutations of Phe453 and Leu328 which were included in the first version of this work. In sum, they establish that the exquisite specificity of the compound class for USP30 is facilitated through USP30-specific residues in different areas of the compound binding site.

We next confirmed that inhibition of USP30 by Compound 39 as well as by the here described compound NK036 modulate mitophagy, as assessed by increased ubiquitination on Tom20 and Tom40 upon mitochondrial depolarization. For this experiment, we used the well-established system of HeLa cells expressing YFP-Parkin, which was kindly provided by Richard Youle. To enrich ubiquitinated proteins, we used the high affinity ubiquitin binder OtUBD of the Hochstrasser lab (Berk et al. *Nat Commun* 2020, 11, 2343) in a workflow recently applied by our lab (Wendrich, Gallant et al. *Nat Chem Biol* 2024, online). These data are shown in Fig. 5a.

In line with mass spectrometry data from the Harper lab on induced human neurons (Ordureau et al. *Mol Cell* 2020, 77, 1124), USP30 mainly regulates ubiquitination of Tom complex components. However, diGly mass spectrometry cannot distinguish between mono- and polyubiquitinated sites, and previously used assays with polyubiquitin binding reagents cannot visualise monoubiquitinated proteins (Gersch et al. *Nat Struct Mol Biol* 2017, 24, 920). Our analysis thus goes beyond these previous studies. Consistent with these, we observe effects of USP30 inhibitors on the ubiquitination of Tom complex subunits but not on Mitofusin-2. We visualize regulation of Tom40 polyubiquitination by USP30 for the first time, and we reveal that in this system under depolarizing conditions polyubiquitinated species are most affected by USP30 inhibition whereas monoubiquitinated Tom20 and Tom40 appear at similar or only slightly increased levels.

These additional data validate that NK036 can be used like Compound 39 to interrogate USP30 in a cellular context. The explored compound-resistant mutations in USP30 corroborate the binding mode with full-length, cellular USP30 and offer tightly controlled experiments with the USP30 inhibitors to interrogate mechanisms by which USP30 antagonizes mitophagy. We hope that the reviewers appreciate how these additions have strengthened our manuscript.

Reviewer #1 (Remarks to the Author):

Kazi et al., report the crystal structure of USP30 in complex with a small molecule inhibitor. Despite USP30 being a prime drug target for therapeutics and USP30 inhibitors undergoing clinical trials for Parkinson's disease, the structural bases for USP30 small molecule inhibition are unclear. Some prior information was eluded by O'Brien et al., 2023 (<https://pubmed.ncbi.nlm.nih.gov/37385347/>), and this has been incorporated in the current study design.

The novelty of the manuscript stems from engineering a USP30 construct that is readily crystallisable with a small-molecule ligand, which has so far been difficult. This comes from the author's prior knowledge and expertise on USP30 structure and function including their HDX analysis showing mobile USP30 regions. In the current study, the authors have switched these flexible regions with more 'stable' regions that were grafted onto the USP30 catalytic domain, thus generating a USP30 chimera that can be used for crystallisation. The manuscript has ample data to support that the chimeric protein is stable, sensitive to small molecule inhibitors and binds ubiquitin, and the authors have gone to great lengths to justify their use of this approach. As such the manuscript represents the generation of a highly valuable tool for advancing USP30 small-molecule inhibitor efforts. Analysis of their crystal structure bound to a USP30 inhibitor has also generated insights into druggable pockets of USP DUBs. This is useful in explaining the current small molecule inhibitor mechanism of action and can aid future inhibitor design.

There is a PDB entry (8D1T and related 8D0A) showing the structure of USP30 bound to a covalent small molecule inhibitor and nanobody, although no paper has been published, which highlights the importance of this manuscript. Moreover, the strategy described here will dispense from the use of a nanobody.

While some of the data and approach may appear lacking in novelty (the authors didn't discover new inhibitors or an unexpected inhibitor binding site), on balance, this reviewer thinks that the technical advancement in engineering a new construct and the comprehensive structure analyses merit publication in NSMB. This strategy will be useful to many in the field and can become standard practice for those interested in developing small-molecule inhibitors of USP30 and other DUBs.

We are very grateful to the reviewer for the appreciation of our work, the careful analysis and the supportive conclusion. With regards to the appearance of novelty of our work, we would like to emphasize that the identified cryptic pocket occupied by the sulfonamide portion of the compound is entirely novel. Neither this pocket *within* the thumb domain nor the associated conformational change of the switching loop have been observed for any other DUB. This pocket is significant as it accompanies about half the inhibitor's atoms. Expanding the previous manuscript version, we included additional mutations that establish the critical role of USP30-specific residues also in this pocket and thereby balance the parts focused on the Leu73 binding pocket shared by other DUBs. Please note that these data show that the side chain of Ile154 within this pocket is important for highly potent USP30 inhibition by these compounds, yet this residue was not identified in the binding site model in the above mentioned excellent HDX-MS study by O'Brien et al. We are convinced that these additions strengthen the visibility of our work's novelty, and we appreciate that this unexpected binding site and the associated novelty were explicitly brought up by both reviewers 2 and 3. Moreover, we were glad to see that all reviewers have highlighted the transferability of our chimeric engineering approach to other enzymes of this family.

Minor comments:

-- R-free is on the high side for the USP30-inhibitor structure (26.6%). Can the authors improve this or comment on why this is high?

We agree with the reviewer that R_{free} is on the higher side, being in the 20th percentile of all pdb deposited structures. However, when compared to other structures with similar resolution (considering the anisotropic diffraction cut-offs of 2.75/2.89/3.53 Å), the obtained R_{free} value of this structure is just in the middle of the distribution and thus acceptable. This is also evidenced by the pdb validation output.

We have thoroughly refined our model and extensively explored additional changes but have been unable to reduce the value below 26%.

One reason for this will likely be that the crystallized, engineered USP30 protein still contains several flexible loops which had to be modelled as 7 chain breaks in chain A and 6 breaks in chain B. Both chains contain separate sets of 252 residues of the 317 residues in the protein construct, indicating that approx. 20% of all amino acids could not be built. Residues adjacent to these chain breaks showed increasingly weak density and could in many cases not be retained in the model as they led to strong real space correlation outliers. We have thus been rather conservative with model building. Importantly, no chain breaks are in the vicinity of the compound and all residues of the compound binding site are well resolved in both chains.

A second reason is that the data was obtained from blending data collected on three different crystals. While this led to clearly improved resolution, completeness, and signal intensity, it came with a modest increase in overall R_{merge} (see Supplementary Table 3, from 0.120-0.134 to a final R_{merge} of 0.166).

Together with the resolution being in the above-mentioned range and the relatively high Wilson B of 71/82/161 Å², we think that flexible residues and crystal blending account for the obtained R_{free} . Importantly, the final R_{free} value is in the average range of pdb-deposited structures of similar resolution and the obtained ligand geometry is unambiguous (see comments below).

-- Omit maps after simulated annealing or unbiased Fo-Fc maps (before the ligand was modelled and refined) should be shown in Figures 2B and Ext. Data Fig. 5C

As was brought up also by reviewer 2, we have added figures from an unbiased F_o-F_c omit map to Extended Data Figure 5 for both chains. This map was calculated from the protein geometry of the refinement run after which the ligand was first modelled (i.e. before the ligand was modelled).

We also changed the maps shown in Fig. 2b and Extended Data Fig. 5g to composite omit maps created with simulated annealing from the final protein geometry. Both maps demonstrate the quality of the ligand density in both protein chains within the asymmetric unit.

The clear density is also in line with the (for this resolution) moderate B factor average of all ligand atoms (68.8 Å²), which is similar to the average B factor of the surrounding protein atoms. We hope that the reviewer is convinced about our ligand placement, which is also further supported by the evaluation of the additional point mutations as explained above.

Extended Data Fig. 5e
unbiased F_o-F_c omit map (chain A)

Extended Data Fig. 5f
unbiased F_o-F_c omit map (chain B)

Fig. 2b
Composite omit $2mF_o-DF_c$ map (chain A)

Extended Data Fig. 5g
Composite omit $2mF_o-DF_c$ map (chain B)

-- Can the authors compare USP30 structures with covalent inhibitors (PDBid 8D1T; 8D0A) to their own? Are there major differences / commonalities worth commenting on?

We thank the reviewer for this suggestion and have included in a new Extended Data Fig. 8 superpositions with our NK036-bound geometry. The referred entries were deposited into the pdb by scientists associated with Amgen Inc. and Carmot Therapeutics Inc.. They used the previously described USP30 construct 13 (in this manuscript termed construct 1) of Gersch et al. *Nat Struct Mol Biol* 2017 which was stabilized by a custom antibody. However, due to the lack of an associated manuscript, accessory information e.g. on the used antibody or any validation data are lacking. As both entries show very similar geometries around the ligand binding site, we have selected entry 8D1T with the more potent compound (shown also in Extended Data Fig. 1) for comparison.

We have added a paragraph in the manuscript comparing the binding modes of covalent and non-covalent USP30 inhibitors. Importantly, both binding modes are drastically different, with unique conformational changes of the switching loop associated with both. For example, Phe157 moves backwards upon binding of the non-covalent compound (Fig. 3), leading to the formation of the cryptic pocket described in this manuscript. In contrast, Phe157 moves forward upon binding of the covalent compounds, creating a shielded tunnel between the modified catalytic cysteine and the S1 Ubiquitin site. Moreover, upon binding the covalent compound, the Leu73 Ubiquitin binding pocket is closed by Leu328 of USP30 rotating 90° and capping its entry. This contributes to the hydrophobicity of the tunnel, which is occupied by the central portion of the covalent compounds.

This comparison shows that both binding modes are very distinct, and the obtained data are thus highly complementary. We envisage that a more detailed comparison of these binding modes and different compound characteristics will be highly informative. While the compound potency could be inferred from a patent, regrettably, there is no specificity data available for these compounds.

-- How do USP30 (ch3) crystal structures compare to an AlphaFold 3 predicted model? What is the RMSD? Will be useful to compare these and perhaps comment on the ability of AF3 to predict chimeric proteins.

We thank the reviewer for this curiosity and suggestion. We have clarified in the manuscript that all calculations were carried out with AlphaFold 2 (version 2.3, also brought up by reviewer 2). In response to this comment, we have also carried out a prediction of the crystallized construct with AlphaFold 3 and summarize below the obtained C_{α} RMSD values, as mean \pm s.d. of the RMSD of the five output models compared with the experimentally determined geometries:

	USP30^{ch3}~Ub-PA (using approx. 1691 atoms)	USP30^{ch3} + NK036, chain A (using approx. 1375 atoms)	USP30^{ch3} + NK036, chain B (using approx. 1354 atoms)
AlphaFold 2	0.77 \pm 0.09	0.65 \pm 0.07	0.60 \pm 0.06
AlphaFold 3	0.69 \pm 0.06	0.57 \pm 0.02	0.56 \pm 0.03

These numbers indicate that both AlphaFold versions performed exceptionally well on the chimeric sequence and, judging from our limited experience, should be well suitable to predict chimeric proteins.

Interestingly, superpositions of the predicted USP30 geometries with the observed ligand-bound geometry highlight that the only region of graver disparity is the switching loop. This is in line with our conclusion of its compound-induced de novo conformation.

Notably, we also attempted to predict the ligand geometry using the Rosetta TTAfold-All-Atom and the Chai-1 algorithms. However, both algorithms failed to provide outputs that were resembling the experimental structure.

Reviewer #2 (Remarks to the Author):

USP30 is a member of a Ub-specific protease (USP) family and is localized on mitochondria. USP30 regulates the basal level of mitochondrial ubiquitination and opposes Parkin/Pink1-mediated mitophagy, depending on its deubiquitinating activity. The inhibition of USP30 promotes the clearance of damaged mitochondria. Therefore, inhibitors of USP30 have the potential to treat diseases caused by dysfunctional mitochondria, like Parkinson's disease. To date, several scaffolds of small molecule USP30 inhibitors have been reported, but the structural basis for their inhibition mechanisms remains unknown. In this paper, Haque Kazi et al. determined the crystal structure of the engineered human USP30 in complex with a specific non-covalent inhibitor. The structure and structure-based site-directed mutagenesis study elucidated how this compound specifically inhibits USP30 at the atomic level. The chimeric engineering strategy for improving the crystallizability of USP family enzymes may be useful for structure-based drug development targeting this enzyme family. Overall, every experiment was appropriately performed, and the result is explicitly presented, although there is a concern regarding the diffraction anisotropy of the inhibitor-bound structure. The findings including an identification of the unexplored cryptic ligand binding pocket are novel and may attract considerable interest to readers, particularly in the research field for the biological system associated with ubiquitin and autophagy. I suggest the following points to be addressed before publication.

We thank the reviewer very much for the praise and the careful description of our work.

Major points:

1. Please provide the statistical information of the inner shell (i.e., at the resolution limit along the lowest resolution axis) in the diffraction anisotropy analysis by STARANISO. Spherical data completeness in the inner shell is important to assess the data quality.

We agree with the reviewer that data completeness in the inner shell is an important indicator of overall data quality, which is not included in the conventional "Table 1" format in publications. We here provide

the summary of merging statistics of the STARANISO output, with the requested statistical information of the inner shell listed in the middle column:

	USP30 ^{ch3} + NK036 (PDB: 9F19)		
	Overall	Inner Shell	Outer Shell
Low resolution limit (Å)	59.522	59.522	3.226
High resolution limit (Å)	2.748	6.107	2.748
R_{merge}	0.166	0.078	1.536
R_{meas}	0.173	0.081	1.595
R_{pim}	0.046	0.022	0.426
Total number of observations	214,419	29,849	30,662
Total number unique	15,438	2,207	2,202
Mean(I) / $\sigma(I)$	9.6	21.2	1.9
Completeness (spherical) (%)	68.7	100.0	26.2
Completeness (ellipsoidal) (%)	91.9	100.0	69.0
Multiplicity	13.9	13.5	13.9
$CC(1/2)$	0.998	0.998	0.730

To complement Supplementary Table 3 and enable further assessment of data quality, we here also provide statistical information of the inner shells of the three individual datasets which were merged into the data used for the STARANISO analysis:

	Crystal 1 Inner Shell	Crystal 2 Inner Shell	Crystal 3 Inner Shell
Low resolution limit (Å)	48.887	68.713	53.890
High resolution limit (Å)	8.312	8.982	7.837
R_{merge}	0.068	0.035	0.049
R_{meas}	0.078	0.040	0.056
R_{pim}	0.037	0.018	0.026
Total number of observations	3,748	2,946	4,841
Total number unique	848	654	1,050
Mean(I) / $\sigma(I)$	17.6	20.7	20.1
Completeness (spherical) (%)	93.4	88.9	96.2
Completeness (ellipsoidal) (%)	93.4	88.9	96.2
Multiplicity	4.4	4.5	4.6
$CC(1/2)$	0.939	0.999	0.944

While for all individual data sets we used only the first 1000-1500 frames to reduce the impact of radiation damage, the blended data achieves a very high spherical completeness in the inner shell (100.0%). Moreover, all datasets (both the individual ones and the blended one) display identical completeness values for spherical and ellipsoidal resolution cut-offs in their inner shells.

We hope that the reviewer is satisfied with this information and agrees that there are no issues with these data.

Also, please transform the crystal lattice axis so that the space group becomes P2₁2₁2 if possible. P2₁2₁2₁ may be unconventional for protein crystallography.

We recognise that the default setting for space group 18 is defined with the unique axis in c (that is, as P2₁2₁2). However, as is evident from discussions on the CCP4 bulletin board, even lead authors of key crystallography programs such as Aimless/Pointless as well as Dials are rather outspoken about using P2₁2₂1, stating with reference to this group specifically that it is “consistent with international conventions and cause[s] no real confusion”. (Ref: <https://www.jiscmail.ac.uk/cgi-bin/webadmin?A2=ccp4bb:17b5b071.1601>). According to the PDB Statistics page the PDB has 544 entries using space group P2₁2₂1, so this setting appears well accepted by the community (<http://rcsb.org/stats/distribution-space-group>).

In our case, only the presence of one screw axis was unambiguously determined from systematic absences whereas the decision on the other two axes became clear only after successful molecular replacement. While we collected data over multiple synchrotron trips (with ultimately the additive screening allowing suitable resolution), we treated all data according to the $a < b < c$ convention for easy comparison of datasets. As transforming the data now (as suggested by the reviewer) would require re-refinement and re-deposition with the PDB with no additional insight (and as $P2_122_1$ appears to be accepted and “consistent with conventions”, if not even preferred by parts of the community), we kindly refrain from following this suggestion but are taking this advice with us for future work in this crystal form.

However, to ensure that this space group notion did not interfere with data processing (as also stated in the above referenced discussion thread), we re-inspected log files of all programs to confirm the absence of any space group-related error or warning messages. Moreover, we re-indexed the three individual datasets into $P2_12_12$ and repeated the processing (blending, anisotropic scaling/merging, molecular replacement, three rounds of automated refinement with simulated annealing starting with an intermediary geometry). We obtained identical merging statistics and virtually identical refinement results, which confirms that all programs used can process data in $P2_122_1$ and that there is no concern.

2. An F_o-F_c omit map should also be shown for the bound inhibitors. I prefer an F_o-F_c omit map to a $2F_o-F_c$ composite omit map to show the quality of the ligand density.

As was brought up also by reviewer 1 and is described in the response above, we have added figures from an unbiased F_o-F_c omit map to Extended Data Figure 5. These demonstrate the quality of the ligand density in both protein chains within the asymmetric unit. We hope that the reviewer is convinced about our ligand placement, which is also further supported by the evaluation of the additional point mutations as explained above.

In addition, $2F_o-F_c$ density maps in a few regions other than the ligand should also be presented to guarantee the data quality. I am not sure whether the diffraction anisotropy correction worked fine or not from the data presented in the current manuscript.

We agree with the reviewer that a presentation of the $2F_o-F_c$ density map is useful for readers to judge the quality of the data processing. We have thus added two panels to Extended Data Fig. 5 covering the $\alpha 5$ helix in both chains. In addition, we show on the right two further regions ($\alpha 1$ helix in chain A and $\beta 8$ strand in chain B) for the reviewer to judge the quality of the model. As suggested, these cover a few regions and demonstrate good quality of the density for both chains.

Please note that we applied B factor sharpening of -30 \AA^2 to these maps (as implemented in phenix.refine) as we found this to help discern features of the map during model building. Our case of a medium-to-low resolution structure combined with anisotropy and an elevated Wilson B is referenced as a case which can benefit particularly from this strategy as described by Liu and Chong, *J Mol Biol* (2014) 426, 980-993, with -30 \AA^2 being a very conservative setting to avoid over-sharpening. For clarity, we mention the sharpening in both the methods section and the figure caption.

Extended Data Fig. 5c

$2mF_o-DF_c$ map: $\alpha 5$ helix (chain A)

Extended Data Fig. 5d

$2mF_o-DF_c$ map: $\alpha 5$ helix (chain B)

for reviewer response

$2mF_o-DF_c$ map: $\alpha 1$ helix (chain A)

for reviewer response

$2mF_o-DF_c$ map: $\beta 8$ strand (chain B)

To convince the reviewer further about the diffraction anisotropy correction, we also scaled the blended data with isotropic cut-offs with Aimless. Data were either truncated to 2.75 Å (leading to egregious outer shell $\| \sigma(I)$ of 0.3 and R_{merge} of 9.431, with 2.75 Å corresponding to the cut-off applied to the highest resolution axis in the anisotropic processing) or to 3.15 Å, which led to similar statistics than the used ones (outer shell $\| \sigma(I)$ of 1.8, overall R_{merge} of 0.164, outer shell R_{merge} of 1.477), but at the expense of a nominally 0.4 Å lower resolution.

The impact of both the anisotropy correction and of the B factor sharpening were particularly pronounced in the earlier rounds of model building and refinement. They elevated density features particularly around chain breaks and around regions that were not present in the initial model. We hope that these data can convince the reviewer that the diffraction anisotropy correction indeed worked fine and that it facilitated density interpretation.

Minor points:

1. What does "MOMP" indicate in Fig. 1a? Mitochondrial outer membrane proteins?

The reviewer is correct. We apologise for omitting a definition of this abbreviation. We have added the explanation to the figure legend.

2. "AlphaFold-based modeling" may be "AlphaFold2-based modeling". As far as I know, the architecture of AlphaFold2 is substantially different from that of the initial version of AlphaFold.

Thank you, we have indeed used AlphaFold2 for all structure predictions as written explicitly in the methods section and as cited. To avoid misunderstandings, we have added the "2" to all AlphaFold references in the text and changed the AF abbreviations in the figures to AF2.

3. "F" that indicates structure factors should be italicized and the subsequent "o" (meaning "observed") and "c" (meaning "calculated") should be subscripts.

We agree with the reviewer. We have italicized all structure factor references in both text and Figures and confirmed that all associated "o" and "c" are subscripts. We will ensure that this formatting is kept in the final layout of the manuscript.

4. In Fig. 2b, the color of blocking loop 2 is similar to that of other regions and difficult to distinguish.

Thank you. We have changed the colours of the blocking loops in Figure 2d.

Reviewer #3 (Remarks to the Author):

In this study the authors design and evaluate chimeric USP30 constructs in order to increase their ability to obtain co-crystal structures with USP30 inhibitors. Utilising this strategy they are then able to demonstrate for the first time the binding mode of a close analogue of selective reversible USP30 inhibitor 'compound 39', in this case their analogue is NK-036. The co-crystal structures obtained reveal that the para-Fluoro-benzoyl functionality binds deep in a conserved USP pocket (next to Leu73) which when compared to other known DUB inhibitors suggests this to be a highly ligandable binding site from which future Dub inhibitors could be anchored. Intriguingly, they also show that the rest of the molecule binds in a cryptic pocket that is induced via ligand binding. This is an important, well presented and well supported study that ignites future opportunities and new understanding for DUB inhibitor design and

discovery, particular for selective and reversible DUB inhibitor discovery which has been highly challenging. The data is very sound and appears very robust. I would strongly recommend this for publication with only very minor corrections / non-essential suggestions and congratulate the authors on a fantastic piece of work that the community will I'm enjoy reading and find very useful.

We are very grateful to the reviewer for these supportive comments and the strong support of publication.

Minor correction and suggestion:

- Bottom of page 2, line 58, the sentence starting "Two structurally yet undisclosed..." does not make sense/convey the meaning I think the authors intend and needs to be rephrased.

Thank you, we have rephrased the sentence (as also only the kidney disease trial compound has not been publicly confirmed).

- These reversible USP30 inhibitors are not particularly drug-like in nature. The specificity for USP30 is described here as arising from the binding mode within the Leu73 pocket. Although this is shown to be a ligandable site for multiple DUBs, the F435Y switch in particular supports that this is a key driver in specificity over other DUBs. It would therefore be interesting to observe what contribution is made to affinity and specificity of the rest of the molecule (i.e. the sulfonamide portion). i.e. if one takes the fragment of the portion of the molecule that binds in the Leu73 pocket only, is their measurable affinity to this construct and what is the delta versus the full molecule? This may act as an anchoring fragment for future, more druglike reversible USP30 inhibitor design, and support those suggestions put forward by the authors in the discussion.

We agree with the reviewer that the dipeptide-like core may render the molecule not particularly drug-like but would like to point out that data demonstrating issues upon in vivo use are lacking. Compound 39 is one of the most potent and specific DUB inhibitors among all inhibitor classes targeting human DUBs. We thus also agree with the reviewer that our structure provides a path towards a next generation of reversible USP30 inhibitors.

[Redacted text block]

[Redacted text block]

[REDACTED]

[REDACTED]

[REDACTED]